# On-Policy RL Meets Off-Policy Experts: Harmonizing Supervised Fine-Tuning and Reinforcement Learning via Dynamic Weighting

**Wenhao Zhang, Yuexiang Xie, Yuchang Sun, Yanxi Chen, Guoyin Wang, Yaliang Li**[*]**, Bolin Ding, Jingren Zhou**
Alibaba Group

## Abstract

Supervised Fine-Tuning (SFT) and Reinforcement Learning (RL) are two prominent post-training paradigms for refining the capabilities and aligning the behavior of Large Language Models (LLMs). Existing approaches that integrate SFT and RL often face the risk of disrupting established response patterns and inducing overfitting to expert data. To address this, we present a novel investigation into the unified view of SFT and RL through an off-policy versus on-policy lens. We propose CHORD, a framework for **C**ontrollable **H**armonization of **O**n- and Off-Policy **R**einforcement Learning via **D**ynamic Weighting, which reframes SFT not as a separate stage but as a dynamically weighted auxiliary objective within the on-policy RL process. Based on an analysis of off-policy expert data's influence at both holistic and granular levels, we incorporate a dual-control mechanism in CHORD. Specifically, the framework first employs a global coefficient to holistically guide the transition from off-policy imitation to on-policy exploration, and then applies a token-wise weighting function that enables granular learning from the expert, which promotes on-policy exploration and mitigates disruption from off-policy data. We conduct extensive experiments across various practical tasks, providing empirical evidence that CHORD achieves a stable and efficient learning process. By effectively harmonizing off-policy expert data with on-policy exploration, CHORD demonstrates significant improvements over baselines. We release the implementation to inspire further research.

## 1 Introduction

Large Language Models (LLMs) have demonstrated remarkable capabilities in a wide array of applications (Yang et al., 2024b; Zhang et al., 2025a; Mialon et al., 2023; Gao et al., 2024). Such significant progress can be largely attributed to two critical post-tuning paradigms that enhance the performance of LLMs in real-world scenarios, i.e., Supervised Fine-Tuning (SFT) (Taori et al., 2023; Zhou et al., 2023) and Reinforcement Learning (RL) (Ouyang et al., 2022; Shao et al., 2024).

These two paradigms present their pros and cons. SFT relies on high-quality expert trajectories to effectively mimic response patterns, which can be sensitive to the quality and quantity of expert data (Ye et al., 2025; Guha et al., 2025). Recent studies also point out that SFT may struggle to generalize beyond mere memorization (Chu et al., 2025) and is vulnerable to exposure bias (Zhang et al., 2019). In contrast, RL encourages LLMs to actively explore, which enables better generalization through learning from direct feedback on their on-policy generations (Chu et al., 2025; Chen et al., 2025b). However, such explorations can sometimes be inefficient, leading to policy degradation caused by entropy collapse (Yu et al., 2025) or over-exploitation of suboptimal strategies.

A prevalent and straightforward approach for integrating the strengths of SFT and RL while mitigating their weaknesses is the sequential *SFT-then-RL* paradigm (Liu et al., 2025b; Lambert et al., 2024). Intuitively, the expert's reasoning patterns learned in SFT guide the RL exploration beyond local optima, and then the on-policy learning in RL mitigates exposure bias inherent in SFT and prevents

---

[*]Corresponding author. Email to {zwh434786, yaliang.li}@alibaba-inc.com

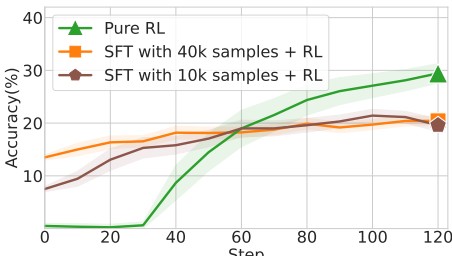
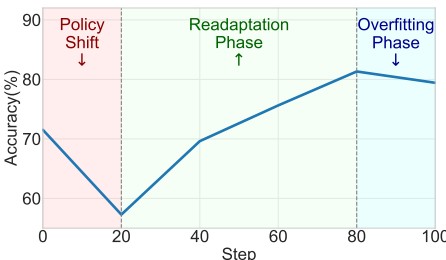

Figure 1: We train Qwen2.5-1.5B-Instruct on the Open-R1 dataset and evaluate the performance on a held-out validation set. These results show that the SFT-then-RL training paradigm can yield suboptimal performance compared to pure RL.

Figure 2: We perform SFT on Qwen2.5-7B-Instruct using expert data generated by Deepseek-R1. The observed learning curve (measured by accuracy on MATH-500) demonstrates a "shift-readapt-overfit" progression.

overfitting to a limited set of static examples. However, empirical observations show that the SFT-then-RL paradigm does not consistently outperform the pure RL approach, as illustrated in Figure 1, which is also noted in recent studies (Zhang et al., 2025a; Chen et al., 2025b).

In this study, we make a further investigation and demonstrate that such suboptimal performance may arise from training on expert data that significantly diverges from the model's established patterns. As illustrated in Figure 2, the learning curve reveals a "shift-readapt-overfit" progression consisting of three distinct phases. Firstly, there is an initial disruption in capability due to the sudden policy shift, which is followed by a readaptation phase during which the model adapts to the expert's patterns and recovers performance. Finally, we observe that the model eventually overfits the expert data. These observations highlight that while expert data can bring new capabilities, it may also *disrupt established patterns and induce overfitting during the training process.*

Drawing upon these insights, we unify SFT and RL through the lens of off-policy versus on-policy learning. The SFT process is reframed not as a separate tuning stage, but as a dynamically weighted auxiliary objective within the on-policy RL process. We further design CHORD, a framework for **C**ontrollable **H**armonization of **O**n- and Off-Policy **R**einforcement Learning via **D**ynamic Weighting. CHORD features a global coefficient $\mu$ for controlling the overall influence of expert data throughout the training process, and a fine-grained weighting function $\phi(\cdot)$ that helps maintain stability via down-weighting highly divergent tokens from off-policy data that could disrupt on-policy training.

Our contributions can be summarized as follows:

- We provide a systematic and in-depth analysis of the training dynamics when employing a separate SFT process to integrate off-policy expert knowledge into models with established policies. We identify the "shift-readapt-overfit" progression, revealing how off-policy data can disrupt the established response patterns of LLMs.

- We propose CHORD, a novel framework that unifies SFT and RL via a dynamically weighted auxiliary loss, which consists of a global coefficient $\mu$ and a token-wise weighting function $\phi(\cdot)$. CHORD provides a fine-grained and flexible control of the influence of off-policy expert data while ensuring training stability, promoting a harmonious integration of learning from both off-policy expert demonstrations and the model's on-policy exploration.

- Extensive experiments demonstrate that CHORD outperforms the SFT-then-RL paradigm and existing approaches. We provide both quantitative and qualitative analyses to show that CHORD strategically navigates training dynamics to selectively absorb expert knowledge without stifling the model's reasoning capabilities, highlighting its superiority and effectiveness.

## 2 PRELIMINARIES

The post-tuning of Large Language Models (LLMs) involves optimizing their policy, denoted by $\pi_\theta$ and parameterized by $\theta$, to generate desirable responses. This typically follows two paradigms: Supervised Fine-Tuning (SFT), an *off-policy* paradigm driven by a static dataset of expert demonstrations; and Reinforcement Learning (RL), an *on-policy* paradigm guided by dynamic feedback.

Specifically, SFT adjusts the policy $\pi_\theta$ to mimic a high-quality, static dataset of $N$ expert demonstrations, $\mathcal{D}_{\text{SFT}} = \{(x_i, y_i^*)\}_{i=1}^N$. Here, $x_i$ is a prompt and $y_i^* = (y_{i,1}^*, \ldots, y_{i,|y_i^*|}^*)$ is the corresponding expert response with $|y_i^*|$ tokens. The SFT objective is to minimize the negative log-likelihood of expert responses, typically optimized with an empirical estimate from a mini-batch of size $B$:

$$\mathcal{L}_{\text{SFT}}(\theta) = -\frac{1}{\sum_{i=1}^B |y_i^*|} \sum_{i=1}^B \sum_{t=1}^{|y_i^*|} \log \pi_\theta(y_{i,t}^* | x_i, y_{i,<t}^*). \tag{1}$$

In contrast, RL optimizes policy $\pi_\theta$ by maximizing expected reward $R(\tau)$ from a generated trajectory $\tau = (x, y^*)$. Group Relative Policy Optimization (GRPO) (Shao et al., 2024) suggests sampling $K$ responses $\{\tau_1, \ldots, \tau_K\}$ from a policy $\pi_{\text{sample}}$ when given a prompt $x$. Each response $\tau_k$ is evaluated with the reward function $R(\tau_k)$, and $\pi_\theta$ is updated to maximize a PPO-style clipped surrogate objective. Consistent with recent studies (Hu et al., 2025a; Yu et al., 2025; Chen et al., 2025a), our formulation does not include the KL divergence term to avoid restricting performance of LLMs. The objective function can be formulated as:

$$\mathcal{L}_{\text{GRPO}}(\theta) = -\frac{1}{\sum_{i=1}^{\hat{B}} \sum_{k=1}^K |\tau_{i,k}|} \sum_{i=1}^{\hat{B}} \sum_{k=1}^K \sum_{t=1}^{|\tau_{i,k}|} \min\left(r_{i,k,t}(\theta) A_{i,k}, \text{clip}(r_{i,k,t}(\theta), 1-\epsilon, 1+\epsilon) A_{i,k}\right), \tag{2}$$

where $\hat{B}$ is the number of prompts in the mini-batch and $\epsilon$ is the clipping hyper-parameter. The advantage $A_k$ for each response is computed by $A_k = \frac{R(\tau_k) - \mu_\mathcal{R}}{\sigma_\mathcal{R} + \epsilon_z}$, where $\mu_\mathcal{R}$ and $\sigma_\mathcal{R}$ are the mean and standard deviation of rewards $\{R(\tau_k)\}_{k=1}^K$ within the group, and $\epsilon_z$ is a small constant for stability. Here $r_{i,k,t}(\theta) \triangleq \frac{\pi_\theta(\tau_{i,k,t}|x,\tau_{i,k,<t})}{\pi_{\text{sample}}(\tau_{i,k,t}|x,\tau_{i,k,<t})}$ denotes the token-wise Importance Sampling (IS) ratio, which re-weights the probability of actions sampled under $\pi_{\text{sample}}$ to simulate on-policy sampled distribution. For a "strict on-policy setup" (Liu et al., 2025b) that $\pi_{\text{sample}} = \pi_\theta$, this ratio should always be 1, and the gradient of $r_{i,k,t}(\theta)$ should be equivalent to $\nabla_\theta \log \pi_\theta(\tau_{i,k,t}^*|x_i, \tau_{i,k,<t}^*)$.

## 3 CHORD: HARMONIZING OFF-POLICY AND ON-POLICY LEARNING

### 3.1 THE SHIFT-READAPT-OVERFIT PROGRESSION WHEN UTILIZING OFF-POLICY DATA

Before introducing CHORD, we first take a close look at the training dynamics of the SFT process, revealing how training on off-policy expert data can disrupt the established response patterns of LLMs. Such disruption ultimately leads to the failure of the SFT-then-RL paradigm (Zhang et al., 2025a; Chen et al., 2025b), as evidenced by the results in Figure 1.

We train Qwen2.5-7B-Instruct (Yang et al., 2024a) on expert data generated by Deepseek-R1 (Guo et al., 2025) and monitor the changes in test accuracy on the MATH-500 dataset. From the experimental results shown in Figure 2, we observe that model performance declines during the first few epochs, followed by a continuous increase to a level higher than that before training, and then a slight subsequent decrease. The performance curve reveals a "shift-readapt-overfit" progression:

- *Policy Shift*: The performance initially declines since the model is forced to follow off-policy expert demonstrations whose response patterns are significantly different, **disrupting its established response patterns and causing a significant performance drop**. Such degradation is further exacerbated by exposure bias (Zhang et al., 2019; Schmidt, 2019), as the model, trained exclusively on ground-truth expert data, struggles to navigate the self-generated contexts it encounters during inference.

- *Readapt*: As SFT continues, the model policy $\pi_\theta$ begins to integrate the expert's response patterns and generates responses similar to those of the expert. The exposure bias can be mitigated by reducing the reliance on the model's response patterns, thereby allowing its performance to rise steadily as it adapts to the expert's response patterns.

- *Overfit*: Extended training on the limited expert data ultimately leads to overfitting, resulting in a decline in generalization and a significant loss of output diversity. Such overfitting can also restrict the exploratory capacity that is crucial for the following RL optimization.

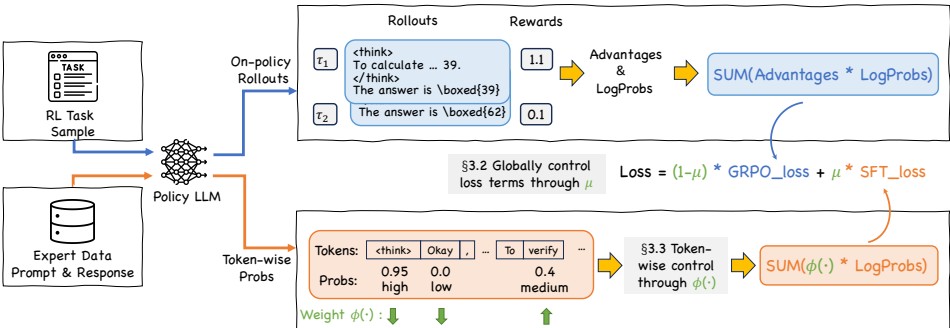

Figure 3: An overview of the proposed CHORD framework that unifies SFT and RL, featuring a global coefficient $\mu$ and a token-wise weighting function $\phi(\cdot)$.

The observed progression makes it challenging to control the influence of off-policy expert data. The SFT-then-RL paradigm demands careful timing for the SFT-to-RL transition, and even then, such a two-stage paradigm may still yield suboptimal solutions due to the inherent separation of the training phases. This highlights the limitations and fragility of the SFT-then-RL paradigm, especially when expert data's response patterns significantly diverge from the model's established response patterns.

Drawing upon the above insights, we propose CHORD, a novel framework that effectively unifies SFT and RL. The proposed framework consists of a dual-control mechanism. We first introduce a dynamic loss coefficient to balance learning from on- and off-policy data (refer to Section 3.2), then further design a token-wise weighting function that provides fine-grained stability control (refer to Section 3.3). The overall architecture of CHORD is shown in Figure 3.

## 3.2 CONTROLLING THE INFLUENCE OF OFF-POLICY EXPERT DATA VIA $\mu$

Firstly, in order to control the influence of off-policy expert data, we propose to reframe SFT as a dynamically weighted auxiliary objective within the on-policy RL process, rather than a separate tuning stage as in the SFT-then-RL paradigm. Specifically, we design a combined loss function that minimizes a weighted sum of the RL and SFT losses:

$$\mathcal{L}_{\text{Hybrid}}(\theta) = (1 - \mu)\mathcal{L}_{\text{GRPO}}(\theta) + \mu\mathcal{L}_{\text{SFT}}(\theta), \tag{3}$$

where $\mathcal{L}_{\text{GRPO}}(\theta)$ is the empirical GRPO loss defined in equation 2, $\mathcal{L}_{\text{SFT}}(\theta)$ is the SFT loss defined in equation 1, and $\mu \in [0, 1]$ is a hyperparameter that governs the trade-off between SFT and RL.

If using a fixed value of $\mu$, the influence of the off-policy expert data remains unchanged throughout the entire post-tuning process. An advanced strategy, however, is to change $\mu$ for achieving a dynamic balance between off-policy and on-policy learning. For example, the SFT-then-RL pipeline can be regarded as a special case with a binary schedule (initially setting $\mu = 1$ and then transitioning to $\mu = 0$). Moreover, previous studies (Ma et al., 2025; Gao et al., 2025) that utilize interleaved SFT and RL can be interpreted as employing a periodic $\mu$ schedule.

Moving a step forward, applying a decay schedule of $\mu$ provides a more graceful and flexible transition from off-policy imitation to on-policy optimization compared to the rigid and binary switch. As shown in Figure 4, the training begins with a large $\mu$ value, encouraging the model to learn more from off-policy expert data. As training progresses, $\mu$ gradually decays to a smaller value, shifting the training focus towards on-policy exploration and annealing the influence of the off-policy expert data before overfitting on them. Such a decay schedule has also proven successful in mitigating exposure bias (Zhang et al., 2019). Inspired by scheduled sampling (Bengio et al., 2015), we generalize the principle of mixing expert and self-generated data from the token level to the loss formulation, effectively bridging the distributional gap between training on off-policy samples and performing on-policy rollouts.

**Beyond the Loss Coefficient** $\mu$    Empirical comparisons (refer to Section 4 for more details) demonstrate that applying a decay schedule to $\mu$ yields notable performance gains over the SFT-then-RL paradigm. At the same time, two key observations motivate us to extend beyond $\mu$.

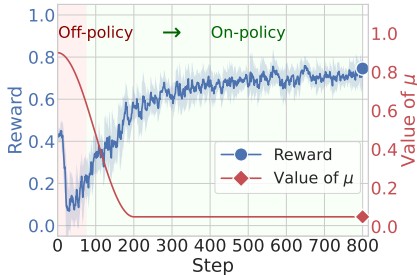
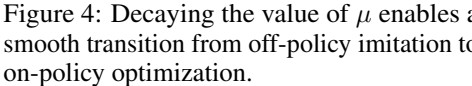
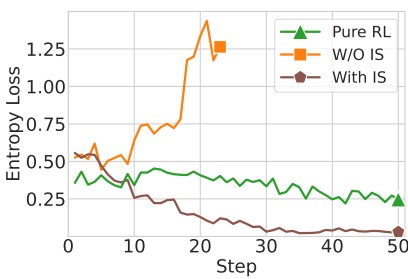

Figure 4: Decaying the value of $\mu$ enables a smooth transition from off-policy imitation to on-policy optimization.

Figure 5: Comparisons of entropy loss between pure RL and mixed RL that integrates expert data (with or without the IS strategy).

Firstly, as shown in Figure 4, the learning curve still reveals a "shift-readapt" progress, where the reward initially declines before subsequently increasing. These observations indicate that, despite improvements in performance, learning from off-policy expert data might still disrupt established patterns and stifle the model's capacity for genuine exploration during on-policy training.

Secondly, the response patterns of the model trained with CHORD-$\mu$ (as shown in Appendix E) appear to converge to those of the expert model. Case studies reveal that CHORD-$\mu$ compels the model to adopt the expert's verbose response pattern wholesale, hence overwriting its own inherent conciseness. This indicates that while $\mu$ controls the overall influence of expert data, it lacks fine-grained precision. As a result, it forces the model to indiscriminately adopt expert patterns, which can create conflicts with its own established style.

Towards the goal of utilizing off-policy data as an incentive and guidance for the model to explore novel and effective reasoning paths, rather than merely as a target to imitate, we further integrate CHORD with a token-wise, fine-grained weighting function $\phi(\cdot)$, forming a dual-control mechanism together with the global coefficient $\mu$ for controlling the influence of the off-policy expert data.

### 3.3 ENHANCING THE STABILITY OF OFF-POLICY LEARNING VIA $\phi(\cdot)$

A feasible solution for controlling the influence of off-policy expert data from a fine-grained perspective is to differentiate the tokens based on their generation probabilities $\pi(y_t^*|x, y_{<t}^*)$. For example, Importance Sampling (IS) (Schulman et al., 2017) has been widely used for stably integrating off-policy data in RL, which suggests re-weighting the objective by the probability ratio between the target policy $\pi_\theta$ and the behavior policy $\pi_{\text{sample}}$ that generated the expert data. Formally, the objective function can be given as:

$$\mathcal{L}_{\text{SFT-IS}}(\theta) = \mathbb{E}_{(x,y^*)\sim\mathcal{D}_{\text{SFT}}}\left[-\sum_{t=1}^{|y^*|}\text{sg}\left(\frac{\pi_\theta(y_t^*|x, y_{<t}^*)}{\pi_{\text{sample}}(y_t^*|x, y_{<t}^*)}\right)\cdot\log\pi_\theta(y_t^*|x, y_{<t}^*)\right], \tag{4}$$

where $\text{sg}(\cdot)$ denotes the stop-gradient operator. Note that the probabilities $\pi_{\text{sample}}(y_t^*|\dots)$ for the expert data $\mathcal{D}_{\text{SFT}}$ are often unknown. Following the common practice (Yan et al., 2025; Wu et al., 2025), we assume that the denominator is 1, treating the expert data as the ground-truth distribution.

From a token-wise perspective, IS enhances training stability by down-weighting low-probability tokens that could disrupt the established policy. As empirical observations shown in Figure 5, mixing off-policy data without IS leads to a sharp rise in entropy, which implies that the model's established patterns are quickly disrupted by the unweighted off-policy data. However, we notice that IS can lead to a sharp collapse in policy entropy compared to pure RL, which implies that it can limit the exploration essential for the RL phase and trap the model in a stable but suboptimal solution. The underlying reason is that IS prevents disruptive shifts in the policy distribution by down-weighting low-probability tokens, but it also aggressively reinforces existing high-probability tokens while ignoring novel but low-probability ones, thus causing the policy to become overconfident.

**Stabilize Off-policy Data Training with $\phi(\cdot)$**    To tackle this, we propose a fine-grained, per-token weighting function $\phi(y_t^*; \pi_\theta)$ that **down-weights the learning signal for tokens at both ends of the probability spectrum**, i.e., down-weighting those tokens that are highly probable (to prevent

entropy collapse) or extremely improbable (to avoid disruption). More specifically, the weight for a given expert token is defined based on the policy's probability $p_t = \pi_\theta(y_t^*|x, y_{<t}^*)$, as follows:

$$\phi(y_t^*; \pi_\theta) = p_t(1 - p_t), \tag{5}$$

which naturally forms a parabolic curve that peaks at $p_t = 0.5$ and decays to zero as $p_t$ approaches 0 or 1. The SFT objective function can be updated as:

$$\mathcal{L}_{\text{SFT-}\phi}(\theta) = -\mathbb{E}_{(x,y^*)\sim\mathcal{D}_{\text{SFT}}}\left[\sum_{t=1}^{|y^*|} \phi(y_t^*; \pi_\theta) \cdot \log \pi_\theta(y_t^*|x, y_{<t}^*)\right], \tag{6}$$

where $\phi(y_t^*; \pi_\theta)$ modulates the gradient contribution of each token in the expert trajectory.

From an information-theoretic perspective, the term $p_t(1 - p_t)$ can be viewed as a measure of the policy's uncertainty (Wang et al., 2025) for the binary event of generating token $y_t^*$. Therefore, this approach biases learning towards tokens where the policy is most uncertain, and creates a "learning sweet spot" that focuses the off-policy learning on tokens that are novel enough to be informative but not so divergent as to disrupt the established policy.

By replacing the static $\mathcal{L}_{\text{SFT}}$ in the proposed hybrid loss function (defined in equation 3) with $\mathcal{L}_{\text{SFT-}\phi}$, we obtain the final objective function of CHORD, which applies a global coefficient $\mu$ for adjusting the overall influence of expert data and a fine-grained weighting function $\phi(\cdot)$ that helps enhance the stability when learning from off-policy data.

## 4 EXPERIMENTS

### 4.1 SETUP

**Datasets, Models, and Evaluations** We conduct experiments on mathematical reasoning problems and practical tool-use tasks. (i) For **mathematical reasoning problems**, we utilize the OpenR1-Math-220k dataset (Hugging Face, 2025), from which we sample 5k instances for SFT and 20k for RL, ensuring no overlap. Our policy model is Qwen2.5-7B-Instruct, whose response patterns differ significantly from the expert (Deepseek-R1). We evaluate in-domain generalization performance on the AIME24, AIME25, and AMC benchmarks (Li et al., 2024), and use MMLU-Pro (Wang et al., 2024) to monitor the changes in general reasoning. (ii) For **tool-use tasks**, we conduct experiments on the single-turn instances of the ToolAce (Liu et al., 2024) dataset. We sample 5k instances for RL and 500 for SFT, for which the expert trajectories are generated by querying the Deepseek-R1 with the same system prompt. We use LLaMA3.2-3B-Instruct (Grattafiori et al., 2024) as our policy model, which also differs in response patterns from the expert (Deepseek-R1). We evaluate the model performance on BFCL (Patil et al., 2024).

**Baselines** We compare the proposed CHORD with a comprehensive set of baselines, including: (i) **Original Model**: The original Qwen2.5-7B-Instruct/LLaMA3.2-3B-Instruct model. (ii) **SFT-only**: The model fine-tuned on the SFT dataset. We focus on two specific configurations: *SFT-light*, trained for a single epoch, and *SFT-best*, the peak-performing checkpoint on the test set found by searching over different learning rates and training epochs. (iii) **RL-only**: The model fine-tuned directly on the RL dataset using the GRPO algorithm. (iv) **SFT+RL**: The sequential SFT-then-RL paradigm. (v) **LUFFY**[1] (Yan et al., 2025): A method that integrates expert demonstrations within GRPO rollout groups and reshapes the importance sampling ratio. (vi) **SASR** (Chen et al., 2025c): A method that probabilistically interleaves SFT and RL steps. It prioritizes SFT when the model's outputs are dissimilar to expert demonstrations, adapting the training focus dynamically.

For more details of the experimental setups, please refer to Appendix A.

### 4.2 COMPARISONS

The proposed approaches implemented based on CHORD include (i) **CHORD-$\mu$**: We employ a decay schedule for the loss coefficient $\mu$ to gradually transition from off-policy to on-policy learning, as

---

[1]For math reasoning problems, we utilize 20k samples for training, whereas the original paper utilizes 45k samples and achieves scores of 50.9 on AMC, 17.7 on AIME24, and 14.8 on AIME25. For tool-use tasks, LUFFY utilizes 5k SFT samples instead of 500.

Table 1: Performance comparisons on reasoning problems and tool-use tasks.

| | Math & General Reasoning Problems | | | | Tool-use Tasks | | |
| --- | --- | --- | --- | --- | --- | --- | --- |
| | AMC | AIME24 | AIME25 | MMLU -Pro | BFCL Live | BFCL Non-live | BFCL Overall |
| Original Model | 43.8 | 11.7 | 6.66 | 24.7 | 50.9 | 39.9 | 46.2 |
| SFT-light | 42.5 | 8.54 | 7.80 | 28.0 | 30.8 | 38.4 | 34.0 |
| SFT-best | 55.9 | 15.8 | 15.2 | 38.4 | 59.2 | 84.2 | 69.8 |
| SFT-light + RL | 52.5 | 11.9 | 11.6 | 44.6 | 68.2 | 89.4 | 77.2 |
| SFT-best + RL | 58.4 | 17.1 | 16.3 | 51.3 | 67.4 | 87.9 | 76.1 |
| SASR | 54.0 | 12.7 | 11.1 | 45.1 | 66.0 | 86.5 | 74.7 |
| CHORD-$\mu$ | 60.8 | 18.1 | **17.9** | 43.3 | 69.4 | 88.6 | 77.6 |
| GRPO (Pure RL) | 52.1 | 13.2 | 8.54 | 45.8 | 68.5 | 88.8 | 77.1 |
| LUFFY | 52.8 | 16.6 | 14.3 | 44.0 | 67.2 | 88.0 | 76.1 |
| CHORD-$\phi$ | **62.5** | **18.2** | 17.2 | **56.2** | **69.9** | **90.2** | **78.5** |

detailed in Section 3.2; and (ii) **CHORD-$\phi$**: We fix the value of $\mu$ and further integrate the token-wise weighting function $\phi(\cdot)$ to achieve a dual-control mechanism on the influence of off-policy expert data, as introduced in Section 3.3.

**Model Performance**   Overall, the comparisons summarized in Table 1 demonstrate the effectiveness and superiority of CHORD on both reasoning problems and tool-use tasks.

Specifically, the experimental results reveal a challenge within the SFT-then-RL paradigm. We notice that minimal tuning on off-policy data (SFT-light) degrades performance, and a more thorough SFT phase (SFT-best) achieves better results. However, the optimal timing for transitioning from SFT to RL can vary across different scenarios. For example, initiating RL from SFT-best yields superior performance on math reasoning problems, while SFT-light+RL performs better on tool-use tasks. This divergence confirms that the SFT-RL balance is highly task-dependent and needs extensive efforts for careful adjustment.

These SFT-then-RL approaches are surpassed by CHORD-$\mu$, which enables a smooth transition from off-policy to on-policy learning rather than a rigid switch. Specifically, CHORD-$\mu$ outperforms the strong SFT-best+RL baseline across all math reasoning benchmarks, achieving improvements of +2.4 on AMC, +1.0 on AIME24, and +1.6 on AIME25, respectively. Besides, CHORD-$\mu$ also achieves better overall results compared to these SFT-then-RL baselines on tool-use tasks. These results demonstrate the superiority of its unified learning design.

Further, CHORD-$\phi$ achieves consistent outperformance over the baselines. These results demonstrate the effectiveness of our dual-control mechanism in flexibly controlling the influence of off-policy expert data. CHORD-$\phi$ selectively applies the SFT loss to non-disruptive tokens, integrating expert knowledge without compromising foundational abilities. This enables robust learning from both off-policy expert data and on-policy exploration, leading to the best performance on both reasoning problems and tool-use tasks.

**Response Patterns**   We further compare the influence of expert data (generated by DeepSeek-R1) on response patterns across different approaches. As shown in Table 2, expert responses are substantially longer than the original model's on both math (6,132 vs. 659 tokens) and tool-use tasks (315 vs. 147 tokens). SFT models (SFT-light and SFT-best) initially mimic this verbosity. However, a subsequent RL can help mitigate the issues of overly lengthy responses by training the models to conduct on-policy exploration. The response length produced by SFT-light+RL is much shorter than that of SFT-best+RL (1,322/119 vs. 4,830/489 tokens), as fewer epochs of SFT allow the model to retain its original response patterns. Besides, from Figure 6, we can observe that CHORD-$\mu$ exhibits a similar trend, where the average response length initially increases to align with expert patterns and then gradually converges to a lower length as on-policy training progresses.

On the other hand, Pure RL on instruct-tuned models lengthens math responses (from 659 to 1,423 tokens) while shortening them for tool-use (from 147 to 118 tokens). This suggests that the response pattern changes can be task-dependent: math problems benefit from detailed step-by-step reasoning, whereas tool-use tasks favor shorter, concise action sequences. Such task-dependent property also

Table 2: Average response length on math problems and tool-use tasks.

|  | Average Length | |
|---|---|---|
|  | *Math* | *Tool-use* |
| Expert Data | 6,132 | 315 |
| Original Model | 659 | 147 |
| SFT-light | 9,966 | 259 |
| SFT-best | 8,442 | 527 |
| SFT-light + RL | 1,322 | 119 |
| SFT-best + RL | 4,830 | 489 |
| CHORD-$\mu$ | 6,081 | 197 |
| Pure RL | 1,423 | 118 |
| CHORD-$\phi$ | 2,444 | 120 |

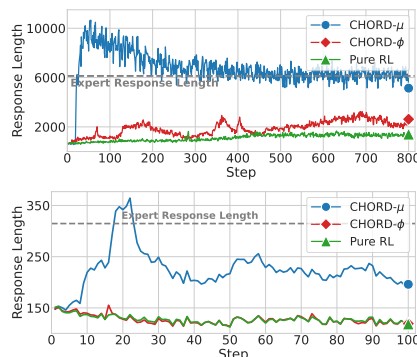

Figure 6: Comparisons of average response length on math problems (top) and tool-use tasks (bottom).

affects the SFT/RL synergy dynamics, as MATH tasks benefit from imitating an expert's long and verbose reasoning via SFT, while such patterns can be detrimental to tool-use performance, a distinction further detailed in Appendix D.2. The result shows that the proposed CHORD-$\phi$ strikes a more nuanced balance: while it also learns to produce more comprehensive mathematical reasoning (2,444 tokens), it generates concise and efficient responses for tool-use tasks (120 tokens). This suggests that the token-wise weighting in CHORD-$\phi$ enables the model to selectively integrate patterns from those of expert data in a task-specific manner. Qualitative analysis shown in Appendix E also confirms the effectiveness of such a flexible design, suggesting that the proposed CHORD-$\phi$ can **go beyond simply mimicking the expert, and learn to selectively absorb reasoning patterns from the expert, while exploring its own response strategies**.

### 4.3 ANALYSIS ON THE EFFECTS OF $\mu$ AND $\phi(\cdot)$

We provide analysis on the effects of the coefficient $\mu$ and the token-wise weighting function $\phi(\cdot)$.

**Dynamic $\mu$ Versus Fixed $\mu$** In Figure 7, we compare the model performance when applying a dynamic schedule for $\mu$ (decreasing from 0.9 to 0.05 over the first 200 training steps and keeping unchanged in the following steps) against several fixed schedules in CHORD. We observe that applying a fixed $\mu$ consistently results in poorer performance compared to dynamic $\mu$. This indicates that naively incorporating off-policy SFT data with a static weight does not effectively serve as a solution for simultaneously learning from off-policy data and on-policy exploration. In fact, it might fail to match Pure RL, which directly encourages an instruction model to follow its own reasoning patterns, highlighting the importance and necessity of controlling the influence of off-policy data.

Besides, while using a smaller value of $\mu$ (e.g., 0.02) can mitigate the performance degradation compared to larger values (e.g., 0.1 and 0.5), it does not provide a significant improvement over pure on-policy RL. With a fixed $\mu$, the model is consistently required to accommodate two potentially divergent reasoning patterns, which might pull it in different directions and prevent it from converging to a stable and high-performance state. The decay schedule for $\mu$ effectively resolves this conflict by creating a smooth transition from off-policy supervision to on-policy exploration.

We also explore a natural extension to adaptively adjust the loss weight $\mu$ based on on-policy rewards. Specifically, the weight is computed as $\mu = \max(0, \tau - \text{reward\_mean})$, which phases out the SFT loss on the expert data as performance improves. The experiment results in Appendix B.1 imply that an automated, reward-aware schedule for $\mu$ can work effectively but requires heavy hyper-parameter tuning, which further motivates the need for fine-grained control.

**Training Curve of CHORD-$\phi$** In Figures 8 and 9, we compare the entropy loss and rewards of Pure RL with those of CHORD-$\phi$ (with fixed $\mu = 0.1$), to illustrate their training dynamics.

From the changes in entropy loss, we can observe that by applying $\phi(\cdot)$, the model maintains a great balance between exploration and exploitation while performing off-policy and on-policy learning simultaneously. On one hand, CHORD-$\phi$ prevents the entropy from collapsing prematurely, which may occur when the SFT loss forces the model to become over-confident on high-probability tokens

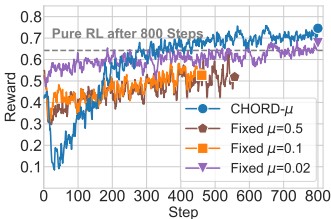 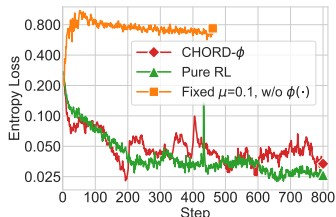 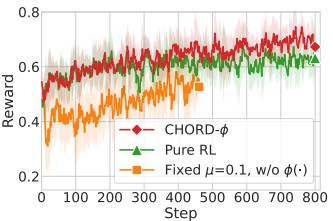

Figure 7: Reward versus training step for CHORD-$\mu$ and various fixed-$\mu$ strategies.

Figure 8: Entropy loss versus training step for CHORD-$\phi$ and baseline methods.

Figure 9: Reward versus training step for CHORD-$\phi$ and baseline methods.

from the expert data. On the other hand, it avoids large entropy spikes and training instability that may occur if the off-policy expert data drastically conflict with the current policy's predictions, as the performance curve remains stable throughout the training process. The rewards curve indicates that CHORD-$\phi$ achieves a stable and continuous increase in rewards, resulting in significantly better performance than Pure RL. These results demonstrate that the proposed token-wise weighting function is crucial for effectively unifying the SFT and RL phases.

**Tuning $\mu$ When Applying CHORD-$\phi$**     Empirical observations show that, when $\phi(\cdot)$ is used for fine-grained control over the influence of expert data, a complex and decaying schedule for $\mu$ is no longer essential. CHORD-$\phi$ is effective to work with a fixed value for $\mu$ (e.g., 0.1 in this study) since it inherently prevents both token-level overfitting and the disruption of established response patterns. The design of $\phi(\cdot)$ simplifies the practical usage of CHORD by making it robust to the specific choice of $\mu$. In Appendix B.7, we provide experiments on tuning the schedule of $\mu$ in conjunction with $\phi(\cdot)$.

**Principle for Instantiating $\phi(\cdot)$**     It is worth noting that the proposed weight $\phi(\cdot) = p_t * (1 - p_t)$ serves as a concrete and interpretable instantiation following a general principle: stabilizing off-policy integration requires down-weighting the learning signal for tokens at both ends of the probability spectrum. This instantiation is computationally efficient, requiring only element-wise multiplication of existing forward-pass probabilities. As grounded in our empirical observations, by assigning negligible weight to tokens that the policy is already certain about (where $p_t$ is close to 0 or 1), the proposed method prevents off-policy data from disrupting the model's established reasoning patterns and focuses updates on tokens where the model is still uncertain. Beyond the specific formulation of $\phi(\cdot)$, this general principle that enables stable and selective learning from off-policy data can potentially inspire more advanced weighting schemes that are suitable for different scenarios.

We also experiment with several variants of the token-weighting function (such as entropy-based variants, clipping variants, and focal loss), with detailed experiment results and discussions presented in Appendix B.2. We acknowledge that this $\phi(\cdot)$ token-wise weighting design primarily illustrates a generalizable principle. While it is unlikely that a single universally optimal design exists across all tasks, datasets, and models, our empirical results confirm that our design nonetheless serves as a highly effective and robust instantiation.

## 4.4 FURTHER ANALYSIS

**Varying Expert Data Source**     We investigate the effect of using two different expert sources: the powerful DeepSeek-R1, and the weaker Qwen2.5-72B-Instruct, whose response pattern is closer to the LLaMA3.2-3B-Instruct policy model. Experimental results demonstrate that our proposed methods, CHORD-$\mu$ and CHORD-$\phi$, outperform Pure RL and SFT+RL baselines regardless of the expert. We also observe that methods which rely more heavily on expert imitation (e.g., SFT+RL and CHORD-$\mu$) can yield greater gains when the expert is stylistically similar to the policy model. This aligns with our insight: the effectiveness of unifying SFT and RL depends not only on expert data quality but also on the degree of pattern shift it introduces. For detailed results and discussion, please refer to Appendix B.3.

**Extending to Non-verifiable Domains**     To test the generalizability of CHORD beyond verifiable tasks, we conduct experiments on RaR-Medicine, a medical question-answering dataset that lacks deterministic verification. The results show that both CHORD-$\mu$ and CHORD-$\phi$ significantly outperform

pure RL. CHORD-$\phi$ achieves faster convergence and higher final rewards, while CHORD-$\mu$ exhibits a similar "shift-readapt" pattern as observed in the main experiments (Figure 10 in Appendix B.4). These findings validate that our approach successfully generalizes to more diverse, non-verifiable domains. Refer to Appendix B.4 for more details.

**Training Weaker Policy Models** While the effectiveness of on-policy exploration (RL) is often limited for weaker models, our experiments reveal that their capability to absorb knowledge from expert data simultaneously degrades. Our experiment on Qwen2.5-3B-Instruct shows that due to this dual limitation, a weaker model tends more to suffer from a performance collapse when training on the same off-policy expert data. In such a setting, naive imitation fails and a simple SFT+RL combination proves unstable, whereas our CHORD-$\phi$ provides a robust objective to navigate this trade-off and maintain learning stability. We acknowledge that with larger quantities of higher-quality expert data, a weaker model might overcome this imitation bottleneck, potentially making SFT-leaning methods more favorable. Conversely, this also suggests that our method becomes particularly advantageous when dealing with weaker models under limited expert data, highlighting its strong capability to flexibly balance this trade-off and create a robust synergy between SFT and RL. We defer detailed experiment results and analysis to Appendix B.5.

## 5 RELATED WORKS

Recent advancements in RL show significant success in complex reasoning tasks (Guo et al., 2025; Shao et al., 2024; Lambert et al., 2024). However, RL-based exploration is often constrained by the model's initial knowledge, making it difficult for the model to discover superior reasoning pathways (Yue et al., 2025). Incorporating off-policy expert data into the on-policy RL loop is a promising strategy to address such exploration challenge. Some studies directly mix expert data with self-rollout generations, either through simple dataset mixing (Li & Khashabi, 2025), or mixing expert trajectories into on-policy rollout groups (Yan et al., 2025; Fu et al., 2025), while others use expert data to guide generation (Liu et al., 2025a; Zhang et al., 2025b; Huang et al., 2025). A third category interleaves RL updates with SFT steps on expert data, either on a predefined or adaptive schedule (Chen et al., 2025c), or for challenging examples (Ma et al., 2025). More recently, SRFT (Fu et al., 2025) proposed a unified framework that combines data mixing with a sample-level SFT loss. In this study, we focus on tuning an instruct model that already establishes its own response pattern, which can be a more challenging yet practical scenario compared to existing works that finetune a base model (Yan et al., 2025; Fu et al., 2025). For a more comprehensive literature review, please refer to Appendix C.

## 6 CONCLUSIONS AND FUTURE DIRECTIONS

In this study, we identify that the sequential SFT-then-RL paradigm often yields suboptimal performance by disrupting established patterns with off-policy expert data. To address this, we unify SFT and RL through the lens of on-policy versus off-policy learning and propose CHORD. By analyzing the influence of expert data at both the holistic and granular levels, CHORD first integrates a global coefficient $\mu$ to manage off-policy expert influence, enabling a smoother transition from imitation to exploration. CHORD then introduces a token-wise function $\phi(\cdot)$ to selectively absorb expert knowledge by down-weighting tokens that are either already highly probable or extremely improbable. Extensive experiments demonstrate that this unified approach achieves notable improvements over existing sequential pipelines and various baselines.

Looking forward, our work highlights several avenues for future research. First, given the inherent high variance and parameter sensitivity typical of RL processes, exploring further algorithmic advancements to fully stabilize the training dynamics represents a promising direction. Specifically within our framework, since configurations for $\mu$ and $\phi(\cdot)$ can vary across setups, discovering more effective adaptive tuning functions to reduce hyperparameter dependence remains a practical objective. Furthermore, while our analysis of pattern shifts is primarily empirical, a deeper understanding of how diverse CoT patterns influence learning would provide valuable insights for tracking training dynamics. Finally, we envision extending this unified paradigm to incorporate heterogeneous mixtures of expert data sources, enabling a comprehensive study of how different model-generated reasoning patterns emerge and evolve throughout the training process. We hope our work inspires the community to continue refining unified post-training paradigms across a broader spectrum of applications.

ACKNOWLEDGEMENTS

We would like to thank Qi Liu, Zexi Li, and Ao Li for their suggestions. We also thank the anonymous reviewers and Area Chairs for their constructive feedback during the review process.

REPRODUCIBILITY STATEMENT

We release our implementation to inspire further research.

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

# A EXPERIMENTAL SETUPS

## A.1 HYPERPARAMETERS

Across all experiments, we adopt the Adam optimizer with $\beta_1 = 0.9$, $\beta_2 = 0.999$. The learning rate is tuned within $\{1 \times 10^{-6}, 5 \times 10^{-6}, 1 \times 10^{-5}\}$, and the temperature for both rollout and evaluation is 1.0. The max response length is set to 16k tokens. For SFT, we train for a maximum of 3 epochs. For RL, we employ "strict on-policy training" similar to (Liu et al., 2025b), where we generate $K = 8$ rollouts per prompt before each policy update.

For mathematical reasoning problems, the batch size for SFR/RL is 64/32, and the maximum number of RL steps is 1,500. For tool-use tasks, the batch size is 96 for both RL and SFT, and the maximum number of RL steps is 100. The $\mu$ decay schedule is to decrease from 0.9 to 0.05 over the first 30 training steps.

## A.2 IMPLEMENTATION DETAILS

In our experiments, the reward function is tailored to the task-specific requirements. For mathematical reasoning problems, we use a hierarchical reward scheme to encourage both correctness and format adherence. To guarantee the precision of our correctness evaluation, we exclusively sample problems that have integer answers when preparing our dataset. A response receives a reward of $+1.0$ for a correct final answer. If the format is correct (e.g., step-by-step reasoning ending with a boxed answer) but the answer is wrong, it receives a neutral reward of $0.0$. A small penalty of $-0.1$ is applied for responses that are both factually incorrect and improperly formatted. Finally, we penalize overly long and inconclusive responses (Yu et al., 2025), and apply a strong penalty of $-1.0$ for exceeding the predefined token limit without a final answer. For tool-use tasks, we employ a simpler binary reward. A response is given a reward of $+1.0$ if it is completely correct, and $0.0$ otherwise.

We implement SFT algorithms based on LLaMA-Factory (Zheng et al., 2024), and implement RL algorithms based on Trinity-RFT (Pan et al., 2025). Experiments are conducted on 8 NVIDIA A100 GPUs and 8 NVIDIA H20 GPUs.

For evaluation, we adopt accuracy as the metric. To avoid high variance in results and ensure fair comparisons, we report avg@32 on AIME24 and AIME 25, and avg@8 on AMC, respectively. Reported results are on the best checkpoint determined by the validation set.

## A.3 PROMPTS

**Prompt for Math Problems** The adopted prompt for math problems is shown below.

---
**Example: Prompt for Math Problems**

```
<|im_start|>system
You are a helpful assistant that solves MATH problems. You should first think about
the reasoning process in mind and then provide the user with the answer. You should
present your reasoning process using the format: <think>\n...your reasoning process
here... </think>\n first. You should always include your final answer in \boxed{} as
closed-form results.<|im_end|>
<|im_start|>user
1. A bus leaves the station at exactly 7:43 a.m. and arrives at its destination at exactly 8:22
a.m. on the same day. How long, in minutes, was the bus trip?<|im_end|>
<|im_start|>assistant
```
---

For the performance of the base model, we report the higher score achieved using either the above prompts for math problems or the default prompt provided by Qwen (Yang et al., 2024b): "Please reason step by step, and put your final answer within \boxed{}".

**Prompt for the MMLU-Pro Dataset** The adopted prompt for the MMLU-Pro dataset is shown below. We use the same system prompt as for the math problems, except that for multiple-choice questions, we modify the answer format to require the corresponding integer as the response.

---

**Example: Prompt for MMLU-Pro Question**

```
<|im_start|>system
```
You are a helpful assistant that solves MATH problems. You should first think about the reasoning process in mind and then provide the user with the answer. You should present your reasoning process using the format: `<think>`\n...your reasoning process here... `</think>`\n first. You should always include your final answer in `\boxed{}` as closed-form results.`<|im_end|>`
```
<|im_start|>user
```
Let V be the set of all real polynomials $p(x)$. Let transformations $T$, $S$ be defined on V by $T : p(x)- > xp(x)$ and $S : p(x)- > p'(x) = d/dxp(x)$, and interpret $(ST)(p(x))$ as $S(T(p(x)))$. Which of the following is true? Below are multiple choice options. You should answer your choice by selecting the index of the option as a number:
0. $ST + TS$ is the identity map of $V$ onto itself.
1. $TS = 0$
2. $ST = 1$
3. $ST - TS = 0$
4. $ST = T$
5. $ST = 0$
6. $ST = TS$
7. $ST - TS$ is the identity map of $V$ onto itself.
8. $TS = T$
9. $ST = S$ `<|im_end|>`
```
<|im_start|>assistant
```

**Prompt for the Tool-use Tasks** For the tool-use tasks, we follow (Zhang et al., 2025a) to adopt their experimental setup and use the prompt provided in their Figure 8. This prompt is consistently applied to train the LLaMA3.2-3B-Instruct policy model and to generate SFT data with the DeepSeek-R1 expert model.

## B EXPERIMENTAL RESULTS AND ANALYSIS

### B.1 ADAPTIVE TUNING $\mu$

In addition to the fixed decay schedule for $\mu$, we explored an adaptive strategy to dynamically adjust the SFT loss weight based on the model's ongoing performance, as measured by the average reward. We conducted experiments to validate this idea.

On the Tool-use task, we implemented a strategy where $\mu$ is adjusted based on the mean reward of the rollouts. Specifically, for a given reward threshold $\tau$, the new $\mu$ is calculated as $\mu' = \max(0, \tau - \text{reward\_mean})$. This mechanism ensures that as the model's average reward surpasses the threshold, the SFT component is gradually phased out ($\mu' \to 0$), allowing the training to focus purely on RL. We tested this with thresholds $\tau = 0.5$ and $\tau = 0.7$.

Table 3: Performance comparison of Adaptive $\mu$ strategies on ToolACE.

| Method | Live | Non-live | Overall |
|---|---|---|---|
| Original Model | 50.9 | 39.9 | 46.2 |
| SFT-best | 59.2 | 84.2 | 69.8 |
| SFT-best + RL | 67.6 | 87.9 | 76.1 |
| GRPO (Pure RL) | 68.5 | 88.8 | 77.1 |
| CHORD-$\phi$ (Ours) | 69.9 | 90.2 | 78.5 |
| CHORD-$\mu$ (Fixed Schedule) | 69.4 | 88.6 | 77.6 |
| Adaptive $\mu$ ($\tau = 0.5$) | 69.7 | 89.4 | 78.1 |
| Adaptive $\mu$ ($\tau = 0.7$) | 65.9 | 88.6 | 75.6 |

The results, presented in Table 3, show that setting the reward threshold to 0.5 yields a strong overall score of 78.1, which is highly competitive with our main approach using a fixed decay schedule. This indicates that dynamically reducing the SFT contribution as the model improves is a viable and effective strategy.

However, we also found that this configuration is still dependent on task-specific hyperparameter tuning. When we set a higher threshold of 0.7, performance degraded significantly. This is likely because the policy was subjected to excessive SFT even when achieving moderately high rewards, disrupting the optimization process.

These experiments serve as a proof of concept, demonstrating that an automated, reward-aware schedule for $\mu$ can work effectively and represents a logical extension of our core ideas. However, since this method still requires tuning another hyperparameter(the reward threshold), its practical implementation is not necessarily simpler to tune. Therefore, we present this as a preliminary exploration into adaptive mixing coefficients, leaving a more thorough investigation of robust and generalizable adaptive schemes as a promising direction for future work.

### B.2 VARYING THE $\phi$ FUNCTION

To validate the robustness of our approach and explore alternative weighting strategies, we conduct ablation studies comparing different variants of the token-wise weighting function $\phi(\cdot)$ against our proposed method. We evaluate these variants on both the tool-use and mathematical reasoning tasks.

**Evaluated $\phi$ Variants.** We compare the following token-wise weighting strategies:

- **CHORD-$\phi$ (Ours)**: Our proposed method, with $\phi(p) = p \times (1 - p)$.
- **Entropy Top**: Only trains on the top 5% of tokens with the highest entropy, setting $\phi(\cdot) = 1$ for these tokens and $\phi(\cdot) = 0$ for others.
- **Entropy Norm**: Normalizes the SFT loss weights based on entropy magnitude, with $\phi(p_t) \propto H(t)$.
- **IS Clip**: Applies importance sampling correction but clips tokens with $p_t > 0.4$.
- **Focal Loss**: Adapts focal loss (Lin et al., 2017) to the SFT context, giving higher weight to tokens with lower probability: $\phi(p) = (1 - p)^\gamma$.

**Experimental Setup.** For the mathematical reasoning experiments here, we relax the strict on-policy training protocol: we synchronize the policy model every 2 training steps (instead of after each update) and increase the number of rollouts per prompt to 16. Training is conducted for 400 steps. For tool-use tasks, we maintain the same setup as described in Appendix A.

**Results and Analysis.** Table 4 presents the results on the ToolACE benchmark, while Table 5 shows the performance on mathematical reasoning tasks.

Table 4: Performance comparison of different $\phi$ function variants on ToolACE (BFCL benchmark).

| Method | Live | Non-live | Overall |
|---|---|---|---|
| GRPO (Pure RL) | 68.5 | 88.8 | 77.1 |
| CHORD-$\phi$ (Ours) | 69.9 | 90.2 | 78.5 |
| Entropy Top | 69.1 | 89.4 | 77.8 |
| Entropy Norm | 69.6 | 89.4 | 78.0 |
| IS Clip | 66.2 | 89.1 | 75.9 |
| Focal Loss | 65.6 | 84.0 | 73.4 |

The experimental results reveal several interesting patterns across the different weighting strategies:

- **CHORD-$\phi$ (Ours):** Our proposed method achieves consistently strong performance across both tool-use and mathematical reasoning benchmarks, demonstrating its effectiveness and robustness.

Table 5: Performance comparison of different $\phi$ function variants on mathematical reasoning tasks.

| Method | AMC23 | AIME2024 | AIME2025 |
|---|---|---|---|
| GRPO (Pure RL) | 55.0 | 12.6 | 7.3 |
| CHORD-$\phi$ (Ours) | 59.7 | 14.0 | 14.2 |
| Entropy Top | 58.8 | 17.2 | 13.8 |
| Entropy Norm | 52.5 | 15.0 | 9.2 |
| IS Clip | 55.0 | 13.9 | 12.0 |
| Focal Loss | 34.4 | 3.8 | 4.2 |

- **Entropy-based Variants (Entropy Top & Norm):** These methods validate the intuition of focusing on uncertain tokens. **Entropy Top** shows particularly strong performance on AIME2024 (17.2), proving that selectively emphasizing high-entropy tokens can effectively integrate expert knowledge. However, their performance gains are not as consistent as our method across all benchmarks.

- **IS Clip:** This variant, which clips high-probability tokens, shows limited effectiveness and even underperforms the pure RL baseline on the tool-use task. This suggests that simply clipping tokens is not a sufficiently nuanced strategy.

- **Focal Loss:** This strategy, which aggressively up-weights low-probability (high-surprise) tokens, leads to severe training instability and a significant performance collapse on both task types. This confirms our hypothesis that giving excessive weight to tokens the model deems unlikely can disrupt its learned reasoning abilities and lead to overfitting on expert patterns.

These results highlight the importance of fine-grained control in token-wise weighting. While various strategies can provide improvements over pure RL in specific scenarios, the choice of weighting function significantly impacts both training stability and final performance across different task domains. We note that our proposed $\phi(\cdot)$ instantiation represents one effective realization of the general principle of down-weighting tokens at both probability extremes. The varied performance of different variants suggests that there remains room for exploring alternative weighting schemes that may be better suited to specific task characteristics or training scenarios, and we hope these empirical observations can inspire future research in this direction.

## B.3 VARYING EXPERT DATA SOURCE

To further validate the robustness and generalizability of our approach, we conduct additional experiments using expert demonstrations generated by Qwen2.5-72B-Instruct instead of DeepSeek-R1. This setup is particularly interesting because Qwen2.5-72B-Instruct, while being a weaker expert model compared to DeepSeek-R1, produces responses with reasoning patterns that are more aligned with the base LLaMA3.2-3B-Instruct model. This allows us to investigate how the choice of expert data source—and specifically, the degree of **pattern shift** introduced—affects the effectiveness of different training methods.

Table 6 presents the performance comparison on the BFCL benchmark using expert data from both DeepSeek-R1 and Qwen2.5-72B-Instruct. The results lead to several key insights.

When using expert data from Qwen2.5-72B-Instruct, which exhibits reasoning patterns closer to those of LLaMA3.2-3B-Instruct, CHORD-$\mu$ achieves improved performance (78.1 vs. 77.6) compared to using DeepSeek-R1 data. This validates our hypothesis that the distributional shift introduced by expert data is a critical factor. When the expert's reasoning pattern is more compatible with the base model's existing policy, the progressive integration strategy of CHORD-$\mu$ can more effectively leverage this alignment, leading to better final performance.

Despite the weaker quality of Qwen2.5-72B-Instruct compared to DeepSeek-R1, CHORD-$\phi$ maintains strong and consistent performance (78.3 vs. 78.5) across both expert data sources. This demonstrates the robustness of the token-wise weighting mechanism, which allows the model to selectively absorb

Table 6: Performance comparison on BFCL benchmark using different expert data sources.

| Method | Live | Non-live | Overall |
|---|---|---|---|
| GRPO (Pure RL) | 68.5 | 88.8 | 77.1 |
| Original Model | 50.9 | 39.9 | 46.2 |
| CHORD-$\mu$ (DeepSeek-R1) | 69.4 | 88.6 | 77.6 |
| CHORD-$\mu$ (Qwen2.5-72B) | **68.8** | 90.6 | 78.1 |
| CHORD-$\phi$ (DeepSeek-R1) | 69.9 | 90.2 | **78.5** |
| CHORD-$\phi$ (Qwen2.5-72B) | 68.9 | **90.9** | 78.3 |
| SFT-best + RL (DeepSeek-R1) | 67.4 | 87.9 | 76.1 |
| SFT-best + RL (Qwen2.5-72B) | 68.1 | 90.6 | 77.7 |

useful patterns while mitigating the negative effects of distributional mismatch, regardless of the expert's absolute strength or stylistic differences.

Interestingly, the baseline SFT-best + RL method also shows notable improvement when using Qwen2.5-72B-Instruct data (77.7 vs. 76.1). This further corroborates our core motivation: the effectiveness of SFT is not solely determined by the quality of expert demonstrations in isolation, but is also heavily influenced by the degree of **pattern shift** they introduce relative to the base model. A smaller pattern shift, even from a weaker expert, can be more beneficial than a larger shift from a stronger but stylistically divergent expert.

**Pattern Examples**  To provide qualitative insight into these pattern differences, we present example responses from DeepSeek-R1, Qwen2.5-72B-Instruct, and the LLaMA3.2-3B-Instruct model after pure RL training below.

---

**Example: Response from DeepSeek-R1 (Stronger Expert, Larger Pattern Shift)**

`<think>`
Okay, let me figure out how to respond to the user. The user mentioned they have an important fact about the Mars climate that they want to store. The assistant asked for the specific fact, and the user provided it: "Mars has the largest dust storms in the solar system which can last for months and can cover the entire planet."

Now, the assistant's role is to help store this information. Looking at the available functions, there's a storeMemory function that takes a "knowledge" parameter. The user's fact is the knowledge to be stored. So, the correct action is to call storeMemory with the provided fact as the argument. There's no other function needed here since the user just wants to store the information. I need to make sure the parameters are correctly formatted as a JSON object with the "knowledge" key. No other parameters are missing here. So, the tool call should be straightforward.
`</think>`
<tool_call> [{"name": "storeMemory", "arguments": {"knowledge": "Mars has the largest dust storms in the solar system which can last for months and can cover the entire planet."}}] </tool_call> `<|im_end|>`

---

**Example: Response from Qwen2.5-72B-Instruct (Weaker Expert, Smaller Pattern Shift)**

`<think>`
The inquirer wants to know the recent disclosed securities report for Apple Inc. with the company code 'AAPL'. I can use the 'getDisclosedToday' function to fetch this information. However, since the inquirer did not specify date range, I will use the current date as both the start and end strings.
`</think>`
<tool_call> [{"name": "getDisclosedToday", "arguments": {"company_code": "AAPL", "start_string": "2023-04-01", "end_string": "2023-04-30"}}] </tool_call> `<|im_end|>`

> **Example: Response from LLaMA3.2-3B-Instruct (Pure RL trained)**
>
> ```
> <think>
> ```
> To answer the question, I need to use a method to retrieve a list of available phone brands from the available database.
> ```
> </think>
> ```
> <tool_call> [{"name": "Get All Phone Brands", "arguments": {}}] </tool_call>
> ```
> <|im_end|>
> ```

As illustrated in these examples, DeepSeek-R1 produces much more verbose and elaborate reasoning with extensive meta-commentary, while Qwen2.5-72B-Instruct adopts a more concise style that is closer to the direct, structured approach learned by LLaMA3.2-3B-Instruct through pure RL training. This qualitative analysis confirms that Qwen2.5-72B-Instruct introduces a smaller pattern shift, which aligns with the quantitative improvements observed in Table 6.

### B.4 NON-VERIFIABLE TASKS

To further test the generalizability of CHORD beyond verifiable tasks, we conduct additional experiments on the RaR-Medicine dataset (Gunjal et al., 2025), a medical question-answering task that requires reasoning and explanation without deterministic verification.

We perform training on the Qwen2.5-7B-Instruct model for 200 steps, with both SFT and RL batch sizes of 96, 8 rollouts per prompt, a learning rate of $1 \times 10^{-6}$, and the $\mu$ decay step is set to 50. We use Qwen3-30B-3A-Instruct as the judge model. The expert demonstrations are sourced from the English subset of the medical-o1-reasoning dataset (Chen et al., 2025d), which contains high-quality reasoning traces for medical questions.

Table 7 and Figure 10 present the experimental results. Both CHORD-$\mu$ and CHORD-$\phi$ significantly outperform pure RL, achieving testset scores of 80.6 and 81.3 compared to 76.8 for pure RL. Figure 10 further show that CHORD-$\phi$ achieves faster convergence and higher final rewards compared to pure RL, indicating more efficient exploration guided by expert demonstrations, where CHORD-$\mu$ possesses a similar "shift-readapt" pattern similar to the main experiment. These results validate that our approach can further generalize to more diverse post-training domains.

Table 7: Comparison of test score and response length on RaR-Medicine task.

|  | *Score* | *Length* |
|---|---|---|
| Expert Data | - | 550 |
| Original Model | 67.5 | 402 |
| Pure RL | 76.8 | 685 |
| CHORD-$\mu$ | 80.6 | 1395 |
| CHORD-$\phi$ | 81.3 | 1128 |

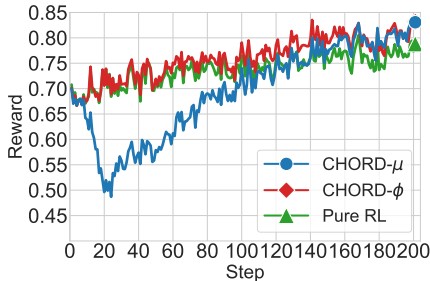

Figure 10: Reward curves for training on the RaR-Medicine dataset.

### B.5 SFT/RL SYNERGY FOR WEAKER POLICY MODELS

An important consideration is how the SFT/RL synergy works when the initial policy model is less capable. Intuitively, one might assume that for a weaker model, supervised fine-tuning (SFT) on expert data would become more critical, as the model's own on-policy exploration is likely to be less effective.

However, our experiments reveal a more nuanced reality: a weaker model can also struggle to effectively absorb knowledge from expert data. As shown in Table 8, naively fine-tuning the weaker Qwen2.5-3B-Instruct model with SFT leads to a performance collapse(RL settings are similar to Appendix B.2). This is in stark contrast to the result observed when training the Qwen2.5-7B-Instruct

model, whose performance significantly improves with the same 5k SFT samples (e.g., AIME2024 accuracy rising from 11.7% to 15.8%). For the weaker model, while Pure RL still provides a consistent performance lift, a naive SFT+RL combination proves unstable. This instability highlights a dual limitation: a weaker model is constrained not only in its on-policy exploration but also in its capacity to absorb off-policy expert data, making the trade-off between exploration and imitation challenging to navigate. In such a setting, the CHORD-$\phi$ method successfully maintains learning stability by providing a robust objective to balance this trade-off. We acknowledge that with larger quantities of higher-quality expert data, a weaker model might overcome this imitation bottleneck, potentially making SFT-leaning methods more favorable. Conversely, this also suggests that our method becomes particularly advantageous when dealing with weaker models under limited expert data, demonstrating its strong capability to flexibly adapt and create a robust synergy between SFT and RL even when naive imitation fails.

Table 8: Performance on MATH tasks with a weaker policy model (Qwen2.5-3B-Instruct).

| Method | AMC23 | AIME2024 | AIME2025 |
|---|---|---|---|
| Original Model (3B) | 33.1 | 4.8 | 1.6 |
| SFT | 22.5 | 1.9 | 1.0 |
| Pure RL | 39.1 | 7.0 | 2.4 |
| SFT+RL | 36.2 | 6.2 | 3.7 |
| CHORD-$\phi$ (Ours) | **41.9** | **7.9** | **4.0** |

## B.6 DIVERSE MODEL ARCHITECTURES

To assess the broader applicability of our method, we extended our evaluation to a model with a distinct architecture and origin: the Phi-mini-MoE-instruct model (Li et al., 2025) (a light-weight Mixture of Experts (MoE) model with 3.8B total, 1.1B active params). This experiment also tests our method's effectiveness beyond the dense Qwen and LLaMA models.

As shown in Table 9, the MoE model exhibits a similar vulnerability to naive SFT in tool-use tasks, with performance collapsing significantly. In stark contrast, our CHORD-$\phi$ effectively achieves the highest performance and boosts the overall accuracy from 49.5 to 61.6.

These results show that our method is architecture-agnostic, and further demonstrates its effectiveness across diverse model families and architectures.

Table 9: Performance on the Tool-Use task with Phi-mini-MoE-instruct model on tool-use tasks.

| Method | Live | Non-live | Overall |
|---|---|---|---|
| Original Model | 42.3 | 59.2 | 49.5 |
| SFT | 18.2 | 45.0 | 29.6 |
| Pure RL | 51.2 | 69.3 | 58.9 |
| SFT+RL | 44.1 | 63.0 | 52.1 |
| CHORD-$\phi$ (Ours) | **52.1** | **74.5** | **61.6** |

## B.7 TUNING $\mu$ IN CONJUNCTION WITH $\phi$

The proposed CHORD employs a dual-control mechanism: a global coefficient $\mu$ and a token-wise weighting function $\phi(\cdot)$. While this raises the question of their joint scheduling, we find that the fine-grained control from $\phi(\cdot)$ makes the framework more robust to the specific schedule of $\mu$. This innovation alleviates the need for meticulous tuning of the global coefficient, simplifying the practical application of CHORD.

The aggressive decay schedule for $\mu$ (starting from a high value) was designed to manage the "shift-readapt" progression. However, since the weight function $\phi(\cdot)$ also aims to stabilize learning and prevent pattern disruption, such an aggressive start may be unnecessary. A more theoretically aligned approach would be to gently introduce the expert data via a warmup-then-decay (Hu et al., 2024)

schedule for $\mu$ (e.g., warming up from 0 to 0.3 before decaying). This would align with the stabilizing nature of $\phi(\cdot)$.

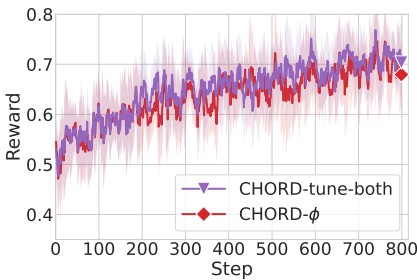

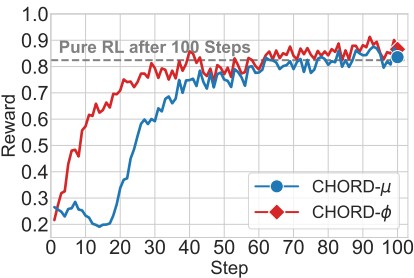

Figure 11: Reward curve comparison for CHORD variants.

Figure 12: Reward curves for training on the ToolAce dataset.

We compare these two schedules in Figure 11. Although CHORD-tune-both that leverages a more refined warmup-then-decay $\mu$ schedule yields a slightly better reward progression during training, the final performance gap between the two approaches is not that significant.

This observation is consistent with our insight: the primary purpose of introducing $\phi(\cdot)$ is to enable expert data to **continuously and stably** guide exploration. By inherently preventing both the disruption of existing patterns and overfitting at a token level, $\phi(\cdot)$ makes the aggressive expert-first approach (a large initial $\mu$) less critical. The token-wise control provides stability, making the overall system less sensitive to the global trade-off hyperparameter. We argue that adopting $\phi(\cdot)$ not only improves stability but also simplifies the practical application of our framework by making it robust to the specific choice of the $\mu$ schedule.

## B.8 EXPERIMENTAL RESULTS ON TOOL-USE TRAINING

We provide the training curves on tool-use tasks in Figure 12 and a more detailed experimental result on the BFCL benchmark in Table 10. The average performance reported in the BFCL benchmark is averaged by instance, meaning that categories with more instances have a greater contribution to the final average score. All methods are evaluated using the same system prompt format.

Table 10: Detailed performance comparisons on BFCL bench.

| | *Live* | | | | *Non-live* | | | | *Overall* | | |
|---|---|---|---|---|---|---|---|---|---|---|---|
| | Simple | Multiple | Parallel | Parallel Multiple | Simple | Multiple | Parallel | Parallel Multiple | Live Avg | Non-live Avg | Overall |
| LLaMA3.2-3B-Instruct | 52.3 | 51.8 | 25.0 | 12.5 | 38.5 | 45.5 | 22.5 | 22.5 | 50.9 | 39.9 | 46.2 |
| SFT-light | 33.7 | 30.8 | 18.8 | 8.3 | 50.5 | 46.0 | 16.0 | 29.0 | 30.8 | 38.4 | 34.0 |
| SFT-best | 69.8 | 57.0 | 68.8 | 37.5 | 77.0 | 89.0 | 77.0 | 76.0 | 59.2 | 84.2 | 69.8 |
| SFT-light + RL | 72.9 | 67.5 | 68.8 | 50.0 | 90.3 | **95.5** | **86.0** | 85.0 | 68.2 | 89.4 | 77.2 |
| SFT-best + RL | 72.9 | 66.1 | **75.0** | 58.3 | **91.5** | 91.5 | 84.5 | 79.0 | 67.4 | 87.9 | 76.1 |
| SASR | 69.4 | 65.3 | 62.5 | 58.3 | 92.0 | 92.0 | 74.0 | 82.5 | 66.0 | 86.5 | 74.7 |
| CHORD-$\mu$ | **74.0** | 68.8 | 68.8 | 50.0 | 83.0 | 92.5 | 83.0 | 84.0 | 69.4 | 88.6 | 77.6 |
| GRPO (Pure RL) | 70.2 | 68.3 | 62.5 | **62.5** | 83.5 | 94.5 | 83.5 | 85.5 | 68.5 | 88.8 | 77.1 |
| CHORD-$\phi$ | 71.3 | **69.8** | 62.5 | **62.5** | 85.0 | 94.5 | 85.0 | **86.0** | **69.9** | **90.2** | **78.5** |

## B.9 EXPERIMENTAL RESULTS ON THE MMLU-PRO DATASET

We provide a more detailed experimental result on the MMLU-pro dataset in Table 11. The adopted prompts for generating these results can be found in Appendix A.3.

# C DETAILED DISCUSSIONS OF RELATED WORKS

## C.1 FINETUNING FOR LLMS

**SFT for LLMs.** SFT has established itself as a cornerstone for aligning LLMs, primarily due to its conceptual simplicity and cost-effectiveness, making it a favored approach within the open-source

Table 11: Detailed performance comparisons on the MMLU-Pro dataset.

| | TAG (by category) | | | | | | | | | | | | | | Average |
| | Business | Law | Psych. | Biology | Chemistry | History | Other | Health | Econ. | Math | Physics | Comp. Sci. | Philosophy | Engineering | Overall Acc. |
|---|---|---|---|---|---|---|---|---|---|---|---|---|---|---|---|
| Qwen2.5-7B-Instruct | 31.18 | 11.72 | 23.81 | 26.22 | 26.15 | 20.73 | 22.40 | 22.74 | 25.95 | 35.75 | 26.48 | 25.12 | 21.84 | 20.02 | 24.71 |
| SFT-light | 40.56 | 8.17 | 21.05 | 25.52 | 36.22 | 14.44 | 23.38 | 24.57 | 27.01 | 44.63 | 37.34 | 28.29 | 17.43 | 21.47 | 28.01 |
| SFT-best | 54.50 | 13.90 | 31.70 | 41.98 | 49.12 | 21.78 | 30.84 | 27.51 | 40.76 | 59.29 | 47.96 | 42.93 | 22.85 | 28.79 | 38.42 |
| SFT-light + RL | 48.80 | 26.52 | 51.50 | 61.09 | 45.41 | 41.21 | 43.72 | 46.82 | 52.73 | 45.89 | 47.19 | 46.10 | 37.68 | 33.95 | 44.61 |
| SFT-best + RL | 60.84 | 26.34 | 51.75 | 64.02 | 56.18 | 40.16 | 49.57 | 49.27 | 57.94 | 62.10 | 57.35 | 51.46 | 43.09 | 39.22 | 51.29 |
| SASR | 52.57 | 23.17 | 47.89 | 59.16 | 46.66 | 36.38 | 44.77 | 42.36 | 55.98 | 52.31 | 51.49 | 46.10 | 36.40 | 30.99 | 45.09 |
| CHORD-$\mu$ | 55.64 | 18.71 | 31.95 | 43.38 | 56.18 | 30.71 | 34.20 | 34.60 | 45.14 | 64.03 | 54.81 | 47.80 | 28.66 | 35.81 | 43.28 |
| GRPO (Pure RL) | 56.91 | 18.35 | 44.74 | 58.58 | 52.30 | 34.38 | 41.23 | 40.22 | 54.86 | 57.88 | 52.19 | 46.10 | 37.07 | 36.02 | 45.77 |
| LUFFY (Yan et al., 2025) | 52.22 | 24.25 | 45.11 | 54.39 | 49.29 | 34.91 | 41.13 | 43.40 | 49.76 | 54.77 | 49.42 | 43.90 | 32.46 | 30.13 | 43.97 |
| CHORD-$\phi$ | 66.79 | 30.88 | 60.78 | 69.87 | 58.30 | 45.93 | 51.19 | 55.13 | 66.35 | 68.47 | 61.66 | 53.41 | 45.89 | 43.14 | 56.22 |

community for creating capable instruction-following models (Taori et al., 2023; Köpf et al., 2023). Early work emphasized the power of high-quality datasets (Zhou et al., 2023; Young et al., 2024), while the required expert curation is labor-intensive and costly. Moreover, to cover the diverse use cases of modern LLMs, the paradigm has shifted towards massive-scale SFT (Grattafiori et al., 2024; Lambert et al., 2024). This trend makes it computationally prohibitive for many to fine-tune from a base model, promoting continued tuning on pre-aligned instruction models instead. Furthermore, the interplay between SFT and RL has grown more complex, from recent methods like DFT (Wu et al., 2025) or iw-SFT (Qin & Springenberg, 2025) that incorporate RL-inspired importance sampling into SFT, to reasoning models like DeepSeek-R1 (Guo et al., 2025) that strategically integrate both paradigms, highlighting that the optimal, principled integration of these methods remains a critical and open area of research.

**RL for LLMs.** Recent applications of Reinforcement Learning (RL) for Large Language Models (LLMs) have expanded beyond traditional human preference alignment (Bai et al., 2022; Ouyang et al., 2022), demonstrating significant progress in complex reasoning domains such as mathematics and code generation (Shao et al., 2024; Yang et al., 2024b; Guo et al., 2025). In particular, a surge of recent work has focused on Reinforcement Learning from Verifiable Rewards (RLVR) (Lambert et al., 2024; Guo et al., 2025), where rewards are derived from definitive outcomes like correct answers or passing unit tests. This paradigm has achieved remarkable results on various benchmarks. However, a fundamental challenge persists in how RL can facilitate effective exploration to surpass the inherent capabilities of its base model (Yue et al., 2025). The search for novel solutions is often constrained by the model's pre-existing knowledge, limiting its discovery of superior reasoning pathways. To address this, introducing external expert data — either for distillation (Hu et al., 2025b; Liu et al., 2025b; Guha et al., 2025), cold start (Guo et al., 2025), or to guide exploration towards diverse, high-quality patterns (Yan et al., 2025; Ma et al., 2025) — emerges as a promising approach to transcend these limitations and unlock new problem-solving frontiers.

## C.2 ON- AND OFF-POLICY REINFORCEMENT LEARNING

**Combining On-policy and Off-policy Data in Traditional RL** In traditional RL domains like robotics (Kober et al., 2013) or games (Mnih et al., 2015), combining on-policy and off-policy data is a potent strategy. Methods ranging from alternating training phases (Gao et al., 2025), to mixing data from separate buffers (Ball et al., 2023), or directly augmenting on-policy replay buffers with expert trajectories (Nachum et al., 2017) have been proven useful. While such methods yield good results in the traditional RL fields, the discrepancy arises from two fundamental distinctions of LLMs: their strong initial priors, where aggressive off-policy updates risk disrupting established reasoning patterns, and their vast, autoregressive action space that radically increases the off-policy degree of expert data, especially for long reasoning chains, and invalidates the assumptions underpinning conventional off-policy algorithms.

**Combining On-policy and Off-policy Data in RL for LLM** Leveraging off-policy data to improve the sample efficiency is a well-established strategy in RL. Several studies have focused on leveraging stale, self-generated data by employing techniques such as refining importance sampling corrections (Tang et al., 2025), mixing on- and off-policy gradients (Li & Khashabi, 2025), modifying the optimization loss objective (Roux et al., 2025; Arnal et al., 2025), or adjusting the synchronization frequency between online and target policies (Lanchantin et al., 2025).

More closely related to our work are methods that leverage external expert data to guide the reinforcement learning process for LLMs. These methods can be broadly categorized. One strategy is direct data mixing (Yan et al., 2025; Dong et al., 2025; Li & Khashabi, 2025). For example, SimpleMix (Li & Khashabi, 2025), operates within a DPO framework and combines off-policy and on-policy data via simple dataset-level sampling. LUFFY (Yan et al., 2025) on the other hand, incorporates off-policy expert trajectories directly into the on-policy rollout groups within a GRPO framework. While such approaches expose the model to expert data, they also introduce significant constraints: they usually require strict prompt alignment between datasets or lack the dynamic, token-level weighting needed to manage severe distribution shifts. Another strategy involves using expert data as guidance for generation. For instance, UFT (Liu et al., 2025a) and BREAD (Zhang et al., 2025b) utilize supervised fine-tuning (SFT) trajectories as prefixes for on-policy rollouts; UFT progressively masks the suffix of the expert demonstration, while BREAD initiates new rollouts by branching from intermediate steps. A third category interleaves RL updates with SFT steps on expert data, either selectively for challenging examples (Ma et al., 2025) or based on a probabilistic schedule (Chen et al., 2025c). Most recently, SRFT (Fu et al., 2025) unifies these approaches into a single-stage framework by not only mixing SFT samples into the on-policy rollout groups but also applying a dedicated SFT loss whose influence is adjusted at the sample level.

Our work diverges from these methods in a crucial aspect. The aforementioned approaches, including state-of-the-art methods like SRFT (Fu et al., 2025), LUFFY (Yan et al., 2025), and Reift (Ma et al., 2025), primarily operate under a "zero-RL" paradigm, initiating training from a base model with a nascent policy. In stark contrast, our work addresses the challenge of fine-tuning a model that already possesses a well-developed, instruction-following policy. This advanced starting point inherently creates a more significant distributional shift between the model's existing policy and the external expert data, thereby exacerbating the off-policy correction problem that our method aims to solve. For further empirical analysis and results, please refer to Appendix D.1.

# D  FURTHER DISCUSSIONS

## D.1  THE INFLUENCE OF OFF-POLICY DATA ON BASE VS. INSTRUCTION MODELS

The challenges of controlling the influence of off-policy data and maintaining training stability are significantly amplified when fine-tuning instruction models. This is mainly due to the established policy inherent in these instruction models.

**Starting from Base Model vs. Instruct Model**    A base model, having been pre-trained solely with a language modeling objective, lacks a coherent, task-specific policy for instruction following. It often has not yet converged on a particular response pattern. When learning from off-policy expert data, the training process is akin to initial policy formation. The model learns a new skill without the risk of conflicting with an existing pattern, thus avoiding significant instability during training.

In contrast, an instruction model has already developed a sharply-peaked policy. Training these models on off-policy expert data that may reflect different reasoning patterns introduces a substantial *distributional mismatch*. The RL algorithm's efforts to reconcile this mismatch can result in large, disruptive policy updates, destabilizing the established policy and potentially leading to a collapse in performance.

Figure 13 provides empirical observation to support the above discussions. When learning from a mixture of on-policy and off-policy data, the reward of a base model improves monotonically, displaying none of the instability issues that can affect instruction models under similar conditions.

Different from most existing studies (Zeng et al., 2025; Yan et al., 2025; Fu et al., 2025), which focus on the "Zero-RL" setting that trains from a base model, this paper addresses a more challenging yet practical problem: how to effectively integrate knowledge from off-policy experts into a model that already possesses an established policy. Training from a base model is not always feasible in practical applications. For instance, such methods are ineffective for tool-use tasks, as the base model typically lacks the basic capability to follow the necessary instructions.

**Applying "Zero-RL" Methods to Our Setting**    To demonstrate the unique advantages of CHORD for aligning **already instruction-tuned** models, we conduct additional experiments comparing our proposed CHORD with LUFFY (Yan et al., 2025) and SRFT (Fu et al., 2025) on the tool-use task.

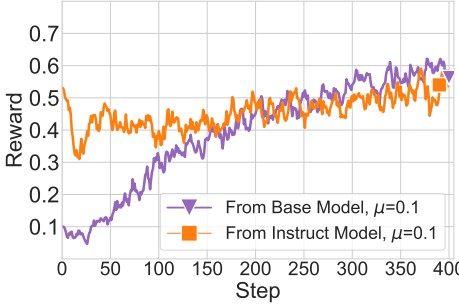

Figure 13: Reward curves for training the base or instruct model with fixed $\mu = 0.1$.

Table 12: Performance comparison with other "Zero-RL" methods on BFCL benchmark. CHORD significantly outperforms "Zero-RL" methods.

| Method | Live | Non-live | Overall |
|---|---|---|---|
| GRPO (Pure RL) | 68.5 | 88.8 | 77.1 |
| CHORD-$\mu$ | 69.4 | 88.6 | 77.6 |
| CHORD-$\phi$ | **69.9** | **90.2** | **78.5** |
| SFT-best | 59.2 | 84.2 | 69.8 |
| SFT-best + RL | 67.4 | 87.9 | 76.1 |
| LUFFY | 67.2 | 88.0 | 76.1 |
| SRFT | 64.6 | 85.8 | 73.6 |

Note that both LUFFY and SRFT require strict alignment between expert demonstrations and RL prompts, as they directly mix expert trajectories into on-policy rollouts. Hence, we generate expert demonstrations for all 5,000 training prompts using DeepSeek-R1. In contrast, CHORD only uses 500 expert demonstrations without requiring prompt-level alignment.

The results in Table 12, show that CHORD significantly outperforms both methods. As discussed in Appendix C, when applied to instruction-tuned models with established policies, directly mixing expert trajectories causes significant distributional mismatch, leading to training instability. Specifically, LUFFY's upweighting of low-probability tokens on top of importance sampling can still cause policy shifts when the distribution gap between expert and policy gap is large. SRFT's uniform sample-level weighting cannot distinguish valuable tokens from irrelevant ones within a trajectory, leading to inefficient and misguided updates. In contrast, our $\phi(\cdot)$ function provides token-wise adaptive weighting, enabling selective absorption of expert patterns while maintaining policy stability. These results validate that our method achieves superior performance with better expert data efficiency and maintains training stability on instruction-tuned models, making it more practical for many more real-world applications.

## D.2 ON DIFFERENT TASK-RELATED PERFORMANCE

The differing performance gains on the MATH and tool-use tasks stem from the fundamental distinctions between these two domains. We deliberately chose these tasks to represent two distinct paradigms, thereby demonstrating the robustness and flexibility of our proposed method.

**The math domain** benefits from complex, structured, and long-form reasoning. For such tasks, acquiring the necessary problem-solving patterns through pure on-policy exploration (i.e., Pure RL) can be inefficient in comparison. Supervised Fine-Tuning (SFT) on expert data is highly beneficial in this context, as it directly exposes the model to well-structured, step-by-step reasoning chains. This allows the model to efficiently learn complex reasoning frameworks that are difficult to discover from scratch. As we discussed in Section 4.2, the model's performance on math problems often correlates with its ability to produce more comprehensive and detailed reasoning steps, a pattern effectively taught by expert data. Therefore, a method that can successfully integrate these expert reasoning patterns, like ours, is expected to yield substantial improvements.

**The tool-use domain**, in contrast, relies more on the exact tool call result rather than the reasoning process. In this setting, naive imitation of expert trajectories through SFT can even be detrimental, as an expert's solution may contain stylistic artifacts (e.g., verbosity) that are not conducive to performance. As shown in Table 2 and discussed in Section 4.2, tool-use tasks favor concise and efficient responses, a pattern that Pure RL naturally learns by shortening response lengths. The primary challenge here is not just to imitate the expert, but to leverage expert guidance to accelerate exploration without being overly constrained or picking up suboptimal habits. The consistent performance gain of our method over the strong Pure RL baseline demonstrates its ability to achieve this delicate balance: successfully extracting useful signals from expert data while avoiding the pitfalls of naive imitation.

These two domains present different challenges for unifying offline SFT and online RL, and our proposed method proves its effectiveness by excelling in both scenarios. It learns to produce comprehensive reasoning for MATH while generating concise, efficient tool calls for tool-use tasks, demonstrating its capability to selectively absorb expert knowledge in a task-specific manner. This validates our approach as a robust and versatile framework for diverse applications.

### D.3    SCALING SFT IS NOT ENOUGH: THE NECESSITY OF ON-POLICY LEARNING

A crucial question is whether extensive SFT on high-quality expert data could eliminate the need for combining SFT and RL. Indeed, as the quantity and diversity of data increase, the problem of exposure bias (Zhang et al., 2019) can be alleviated, leading to better generalization. And for knowledge-intensive tasks like MATH, model performance can be highly correlated with the volume and quality of SFT data. To investigate this, we expanded the MATH SFT dataset from 5k to 20k examples, which substantially boosted the pure SFT model's AIME accuracy from 15% to approximately 24%.

However, even with larger volumes of SFT data, a principled transition to on-policy learning remains critical for reaching the performance frontier. Recent literature (Liu et al., 2025b) also shows that extensive SFT followed by RL fine-tuning is an effective strategy for maximizing model capabilities. By applying our SFT/RL combined approach, we can further elevate the accuracy from 24% to 33%. This demonstrates that RL is not redundant but complementary, enabling the model to refine its policy beyond the static distribution of expert data.

## E    CASE STUDIES

For a better understanding, we compare the generation patterns of RL-only (i.e., pure RL), SFT-only, and the proposed CHORD.

- **RL-only**: The model trained solely with RL exhibits a concise and structured, yet ultimately rigid, reasoning pattern. It tends to follow a fixed template, such as beginning with "To..." and using connectors like "First" and "Next", and proceeds linearly without engaging in self-correction or exploring alternative solution paths. While this approach leads to efficient responses, it may result in less robust solutions.

- **SFT-only and CHORD-$\mu$**: In contrast, the model trained solely on expert demonstrations can be verbose and exploratory. It exhibits a "think-aloud" style with frequent meta-commentary (e.g., "Let me think..."), backtracking, and verification of intermediate steps. The generated responses are comprehensive, but often lack a concise structure. Note that the proposed CHORD-$\mu$ has a similar reasoning pattern to SFT-only.

- **CHORD-$\phi$**: It exhibits a hybrid reasoning style that retains a clean and logical structure while selectively incorporating the expert's sophisticated verification strategies. As shown in the example below, it develops patterns such as "Alternatively..." and "Both methods confirm..." to produce responses that are both well-structured and robust. We select the checkpoints at 800 steps as the models to generate the examples here.

---

**Example: Response generated by Qwen2.5-7B-Instruct**

```
<think>
```
To calculate the duration of the bus trip, we need to subtract the departure time from the arrival time.

The departure time is 7:43 a.m.
The arrival time is 8:22 a.m.
 Subtracting the two times:
- From the hours: 8 - 7 = 1 hour
- From the minutes: 22 - 43 = -21 minutes

---

Since we can't have negative minutes directly, we borrow 1 hour (60 minutes) from the hour part, making it 0 hours and adding 60 minutes to the 22 minutes. Then, 60 + 22 = 82 minutes. So, we subtract 21 minutes from 82 minutes:
82 - 21 = 61 minutes

Therefore, the bus trip duration is 1 hour and 61 minutes, which is simply expressed as 61 minutes.
```
</think>
\boxed{61} <|im_end|>
```

**Example: Response generated by the model trained solely with RL**

```
<think>
```
To determine the duration of the bus trip, we need to calculate the difference in time between the departure time and the arrival time. The departure time is 7:43 a.m. and the arrival time is 8:22 a.m. on the same day.

First, let's identify the departure and arrival times:
- Departure time: 7:43 a.m.
- Arrival time: 8:22 a.m.

Next, we need to find the difference between these two times. We can do this by subtracting the departure time from the arrival time.

From 7:43 a.m. to 8:00 a.m. is the time from 7:43 to 8:00. This is 17 minutes because 8:00 - 7:43 = 17 minutes (since 8:00 is 17 minutes after 7:43).

Now we need to add the time from 8:00 a.m. to 8:22 a.m., which is 22 minutes.

So, the total trip time is 17 minutes (from 7:43 to 8:00) + 22 minutes (from 8:00 to 8:22) = 39 minutes.

Therefore, the duration of the bus trip is `\boxed{39}` minutes. `</think>`
The duration of the bus trip is `\boxed{39}` minutes. `<|im_end|>`

**Example: Response generated by the model trained solely with SFT**

```
<think>
```
Okay, let me try to solve this problem step by step. Let me think aloud so I can explain my reasoning.

So, the problem is: A bus leaves the station at exactly 7:43 a.m. and arrives at its destination at exactly 8:22 a.m. on the same day. How long, in minutes, was the bus trip?

Let me visualize this. Let me think about the time elapsed from 7:43 a.m. to 8:22 a.m.

Let me break this down. Let me think of it as two parts: from 7:43 a.m. to 8:00 a.m., and then from 8:00 a.m. to 8:22 a.m.

Let me check that. Let me see if that makes sense.

So, from 7:43 a.m. to 8:00 a.m., how many minutes is that?

Let me calculate that. Let me think, from 7:43 to 8:00 is 17 minutes. Let me think, let me count:

7:43 to 8:00 is 17 minutes. Because 8:00 minus 7:43 is 17 minutes.

Let me confirm that. Let me think, 7:43 to 7:50 is 7 minutes, and then 7:50 to 8:00 is 10 minutes. So, 7 + 10 = 17 minutes. Okay, that seems right.

So, that's the first part: 17 minutes from 7:43 a.m. to 8:00 a.m.

Then, the second part is from 8:00 a.m. to 8:22 a.m., which is 22 minutes.

So, adding those two parts together: 17 minutes + 22 minutes = 39 minutes.

Let me check that. Let me think, 17 + 22. Let me do the addition:

17 + 22 = 39.

So, that seems correct.

Let me think if there is another way to calculate this.

Let me think, maybe using a different approach.

Let me think, the total time from 7:43 a.m. to 8:22 a.m.

Let me think, let me calculate the difference between 8:22 and 7:43.

Let me break it down hour by hour.

From 7:43 a.m. to 8:00 a.m. is 17 minutes, as we already knew.

Then, from 8:00 a.m. to 8:22 a.m. is 22 minutes.

So, adding those together gives 17 + 22 = 39 minutes.

Let me think if there is a formula or a quicker way.

Let me think, maybe using the formula:

Total time = (End time hour - Start time hour) * 60 + (End time minute - Start time minute)

Let me try that.

So, let me denote the start time as 7:43 a.m.

End time as 8:22 a.m.

So, End time hour = 8

Start time hour = 7

End time minute = 22

Start time minute = 43

So, Total time = (8 - 7) * 60 + (22 - 43)

Let me calculate that.

First, (8 - 7) * 60 = 1 * 60 = 60 minutes

Then, (22 - 43) = -21 minutes

So, Total time = 60 + (-21) = 39 minutes

Oh, that's interesting! So, using this formula, we get 39 minutes as well.

Let me think, let me check if this makes sense.

So, when we subtract the start time from the end time, we get:

8:22 - 7:43

Let me think of this as 8 hours and 22 minutes minus 7 hours and 43 minutes.

But since we are subtracting times, let me think of it as:

8:22 - 7:43 = (8 - 7) hours + (22 - 43) minutes

But since 22 minutes is less than 43 minutes, we need to borrow an hour.

So, 8 hours is 7 hours and 60 minutes.

So, 7 hours and 60 minutes minus 7 hours and 43 minutes is 0 hours and (60 - 43) minutes = 17 minutes.

Then, the remaining hour is 1 hour.

So, total time is 1 hour + 17 minutes = 60 minutes + 17 minutes = 77 minutes.

Wait, but that doesn't match. Let me think if this is correct.

Let me think, maybe this method is complicating it. Let me think, maybe the first method is simpler and more accurate.

Let me think, let me think of it as total time = end time - start time.

So, end time is 8:22 a.m.

Start time is 7:43 a.m.

So, let me calculate the difference.

From 7:43 to 8:00 is 17 minutes.

From 8:00 to 8:22 is 22 minutes.

So, 17 + 22 = 39 minutes.

So, that seems correct.

Let me think, let me think of it another way.

Let me think, let me convert both times to minutes since midnight and subtract them.

Let me think, that might be complicated, but let me try.

Let me think, 7:43 a.m. is 7 hours and 43 minutes.

So, in minutes, that is 7 * 60 + 43 = 420 + 43 = 463 minutes since midnight.

Similarly, 8:22 a.m. is 8 hours and 22 minutes, which is 8 * 60 + 22 = 480 + 22 = 502 minutes since midnight.

So, the difference is 502 - 463 = 39 minutes.

Oh, so that also gives 39 minutes.

So, this method also confirms that the answer is 39 minutes.

Therefore, I think all methods lead to the same answer, so I can be confident that the bus trip lasted 39 minutes.

**Final Answer**
The bus trip lasted \boxed{39} minutes.
</think>

To determine the duration of the bus trip, we need to calculate the time difference between the departure time (7:43 a.m.) and the arrival time (8:22 a.m.).

1. Calculate the time from 7:43 a.m. to 8:00 a.m.:
- From 7:43 a.m. to 8:00 a.m. is 17 minutes.

2. Calculate the time from 8:00 a.m. to 8:22 a.m.:
- From 8:00 a.m. to 8:22 a.m. is 22 minutes.

3. Add the two intervals together:
- Total time = 17 minutes + 22 minutes = 39 minutes.

Thus, the bus trip lasted (\boxed{39}) minutes.<|im_end|>

---

**Example: Response generated by the model trained with CHORD-$\mu$**

```
<think>
```
Okay, let me try to solve this problem step by step. Let me think aloud so I can explain my reasoning.

So, the problem is: A bus leaves the station at exactly 7:43 a.m. and arrives at its destination at exactly 8:22 a.m. on the same day. How long, in minutes, was the bus trip?

Let me visualize this. I need to find the time elapsed between 7:43 a.m. and 8:22 a.m.

Let me break this down. A good way to handle time calculations that cross an hour is to use the top of the hour as a stepping stone. In this case, that's 8:00 a.m. So, I can think of the trip in two parts: from 7:43 a.m. to 8:00 a.m., and then from 8:00 a.m. to 8:22 a.m.

Let me check that. Let me see if that makes sense. Yes, that splits the problem into two easier calculations.

So, first part: from 7:43 a.m. to 8:00 a.m. How many minutes is that?
Let me calculate that. An hour has 60 minutes. So, the time from 7:43 to 8:00 is `60 - 43` minutes.

Let me think... `60 - 40` is 20, so `60 - 43` is 17. Yes, 17 minutes.
So, the first part is 17 minutes.

Then, the second part is from 8:00 a.m. to 8:22 a.m. That's straightforward, it's just 22 minutes.

So, to get the total time, I just need to add those two parts together: `17 minutes + 22 minutes`.
Let me do the addition: `17 + 22`.
`10 + 20` is 30, and `7 + 2` is 9. So `30 + 9 = 39`.
The total is 39 minutes. So, that seems correct.

Let me think if there is another way to calculate this to be sure.
Let me try converting both times to minutes past midnight and then subtracting.

Let me think, start time is 7:43 a.m.
In minutes, that is `(7 hours * 60 minutes/hour) + 43 minutes`.
`7 * 60 = 420`.
`420 + 43 = 463` minutes since midnight.

Now the end time: 8:22 a.m.
In minutes, that is `(8 hours * 60 minutes/hour) + 22 minutes`.
`8 * 60 = 480`.
`480 + 22 = 502` minutes since midnight.

So, the difference is `502 - 463`.
Let me calculate that. `502 - 400` is 102. `102 - 60` is 42. `42 - 3` is 39.
So, `502 - 463 = 39` minutes.

Oh, so that also gives 39 minutes. This method confirms my previous answer.

Let me think about one more method, direct subtraction with borrowing.
I want to calculate 8:22 - 7:43.
Let me write it as hours and minutes.
`8 hours 22 minutes`
`- 7 hours 43 minutes`
I can't subtract 43 from 22. So I need to borrow 1 hour (which is 60 minutes) from the 8 hours.
So, 8 hours and 22 minutes becomes 7 hours and `(60 + 22)` minutes, which is 7 hours and 82 minutes.
Now let me subtract:
`7 hours 82 minutes`
`- 7 hours 43 minutes`
`-------------`
`0 hours 39 minutes`
So, the result is 39 minutes.

All three methods lead to the same answer, so I can be confident that the bus trip lasted 39 minutes.

**Final Answer**
The bus trip lasted (\boxed{39}) minutes.
</think>
To determine the duration of the bus trip, we need to calculate the time difference between the departure time (7:43 a.m.) and the arrival time (8:22 a.m.).

We can solve this by breaking the duration into two parts using 8:00 a.m. as a reference point.

1. Calculate the time from 7:43 a.m. to 8:00 a.m.:
   - From 7:43 a.m. to 8:00 a.m. is `60 - 43 = 17` minutes.

2. Calculate the time from 8:00 a.m. to 8:22 a.m.:
   - From 8:00 a.m. to 8:22 a.m. is 22 minutes.

3. Add the two intervals together:
   - Total time = 17 minutes + 22 minutes = 39 minutes.

Thus, the bus trip lasted `(\boxed{39})` minutes.`<|im_end|>`

---

**Example: Response generated by the model trained with CHORD-$\phi$**

`<think>`
To determine the duration of the bus trip, I need to calculate the time difference between the departure and arrival times. The departure time is 7:43 a.m. and the arrival time is 8:22 a.m. on the same day. The key is to find the difference between these two times.

First, I'll visualize the time on a clock face. Starting at 7:43 a.m., the bus arrives at 8:22 a.m. The difference between these two times is 39 minutes (from 7:43 to 8:22).

Alternatively, I can break down the calculation step by step:

1. From 7:43 a.m. to 8:00 a.m. is 17 minutes (since 8:00 - 7:43 = 0:17).
2. From 8:00 a.m. to 8:22 a.m. is 22 minutes.
3. Adding these two intervals together: 17 minutes + 22 minutes = 39 minutes.

Therefore, the total duration of the bus trip is 39 minutes. Let me confirm this with a different approach to ensure accuracy.

Another way to think about it is to calculate the total minutes between 7:43 a.m. and 8:22 a.m.:

1. Convert 7:43 a.m. to minutes since midnight: 7 hours * 60 minutes/hour + 43 minutes = 420 minutes + 43 minutes = 463 minutes.
2. Convert 8:22 a.m. to minutes since midnight: 8 hours * 60 minutes/hour + 22 minutes = 480 minutes + 22 minutes = 502 minutes.
3. The difference in minutes between 502 minutes and 463 minutes is 502 - 463 = 39 minutes.

Both methods confirm that the bus trip lasted 39 minutes. Therefore, the answer is `\boxed{39}`.
`</think>`
To determine the duration of the bus trip from departure at 7:43 a.m. to arrival at 8:22 a.m., we need to calculate the time difference between these two times.

1. First, let's break down the time from 7:43 a.m. to 8:00 a.m.:
- From 7:43 a.m. to 8:00 a.m. is 17 minutes (since 8:00 - 7:43 = 0:17).

2. Next, calculate the time from 8:00 a.m. to 8:22 a.m.:
- From 8:00 a.m. to 8:22 a.m. is 22 minutes.

3. Add these two intervals together:
- 17 minutes + 22 minutes = 39 minutes.

> Alternatively, we can convert the times to minutes since midnight:
> - 7:43 a.m. is 7 hours * 60 minutes/hour + 43 minutes = 463 minutes.
> - 8:22 a.m. is 8 hours * 60 minutes/hour + 22 minutes = 502 minutes.
> - The difference is 502 minutes - 463 minutes = 39 minutes.
>
> Both methods confirm that the bus trip lasted 39 minutes. Therefore, the answer is
> `\boxed{39}.<|im_end|>`

## USE OF LARGE LANGUAGE MODELS

We used large language models only as general-purpose writing assistants, to proofread and correct grammatical errors in this manuscript.

