# OpenReview forum: "On-Policy RL Meets Off-Policy Experts: Harmonizing Supervised Fine-Tuning and Reinforcement Learning via Dynamic Weighting"
_ICLR.cc/2026/Conference — ICLR 2026 Poster_

### Official Review · Reviewer_pEZ4 · 2025-10-21

**Soundness:** 3
**Presentation:** 3
**Contribution:** 2
**Rating:** 6
**Confidence:** 4

**Summary:**

This paper investigates the interplay between Supervised Fine-Tuning (SFT)  and RL. It proposes a simple yet effective method called Chord, where GRPO and SFT loss are dynamically combined, where the loss computation of SFT is further adjusted based on each token’s policy probability.

**Strengths:**

The major focus — addressing the shift-readapt-overfit phenomenon — is well motivated, and the analysis is insightful. Overall the paper has conducted comprehensive experiments with detailed analysis to explain various design choice.

**Weaknesses:**

The proposed weighting mechanism is not very novel, and the empirical effectiveness is mostly observed on Math.

**Questions:**

1. I don’t see much difference btw GRPO (pure RL) v.s. proposed method in BFCL, even though 5k instances are used, any intuition why the improvement is very large on AIME but not in tool-use?

2. Continue from Q1, since Math benefit more from longer reasoning chain, chord-$\mu$ could mostly learn from expert data (Deepseek-R1)’s format and tendency to generate long reasoing trace, which greately improve the result. From Table 1, the result seems to confirm about this assumption (comparing SFT-best+RL v.s. CHORD). This makes me question if the model is well SFT-trained, is there still any need for combining RL + SFT.

3. Is there any result on base model Qwen3-8B-base? As newer model has much better performance on math/tool-use domain, I am curious if the method still brings improvement across math/tool-use domain.

4. the method of smoothly combining with SFT should be generalizable to non-verifable task as well (and arguably more useful since SFT data could help constrain exploration space to prefered style/format). Any thoughts on such setting?

---

> ### Author Response · Authors · 2025-11-22
> **Response to Reviewer pEZ4 (1/3)**
>
> We sincerely appreciate your detailed comments and valuable suggestions! We provide the following responses to address your concerns and answer your questions point by point.
>
> > W1 & Q1: I don’t see much difference btw GRPO (pure RL) v.s. proposed method in BFCL, even though 5k instances are used, any intuition why the improvement is very large on AIME but not in tool-use?
>
> We would like to point out that the inherent difficulty of the datasets, stemming from their intrinsic properties, affects the upper limit of the model performance and possible improvement. A direct comparison of absolute gains offers an incomplete picture of the method's effectiveness.
>
> More importantly, the effectiveness of SFT and RL is different between settings. As we discussed in Section 4.2, the math tasks benefit from structured, long-form reasoning, while tool use tasks prefer short but accurate responses. For such a reason, we can notice that applying SFT leads to significant improvements on the MATH dataset, since learning from expert data allows the model to improve structured, long-form reasoning skills that are highly beneficial for solving math problems. On the other hand, we observe that simply applying SFT with lengthy expert demonstrations might even **have a negative impact** on tool-use tasks (77.1 for PureRL but 76.1 for SFT-best + RL), while on-policy exploration is favoured as it enables the model to acquire tool-use capabilities effectively via more precise tool call actions.
>
> The proposed method achieves consistent improvements across these two different tasks when compared to SFT, RL, and SFT+RL, which demonstrates the effectiveness of the proposed method in unifying SFT and RL.
>
> Hope the above responses can address your concerns. We have added the detailed discussions to Section 4.2 and Appendix D.2 in the revised paper. Thank you again!

---

> ### Author Response · Authors · 2025-11-22
> **Response to Reviewer pEZ4 (2/3)**
>
> > Q2: This makes me question if the model is well SFT-trained, is there still any need for combining RL + SFT.
>
> Thank you for your constructive comments.
>
> - **Training on larger amounts of high-quality SFT data can significantly improve model performance, especially for math tasks**. For math tasks, training on high-quality SFT data can notably enhance model accuracy, when we increase the amount of SFT data from 5k to 20k, the performance of the trained model on AIME2024 improves from 15% to around 24%. As the quantity and diversity of data increase, Exposure bias (one of the major shortcomings of SFT) can be alleviated, resulting in better generalization performance. However, obtaining large-scale, high-quality, and diverse datasets is costly and can be difficult in certain scenarios, which motivates further exploration in on-policy methods.
> - **Even with larger volumes of SFT data, combining RL with SFT can still yield further improvements**. By further combining SFT and RL training, the performance is raised to 33\%. Recent studies [Ref1] also demonstrate that extensive SFT followed by RL can further enhance performance.
>
> Hope the above responses can address your concerns regarding the need for combining RL and SFT. We have added the above results and analysis to Appendix D.3 in the revised paper. Thank you again for your constructive comments!
>
> [Ref1] AceReason-Nemotron 1.1: Advancing Math and Code Reasoning through SFT and RL Synergy.
>
>
> > Q3: Is there any result on base model Qwen3-8B-base? As newer model has much better performance on math/tool-use domain, I am curious if the method still brings improvement across math/tool-use domain.
>
> We appreciate the reviewer's insightful question regarding the scalability of our findings to the latest models. We believe that our method, which targets a fundamental mechanism in harmonizing off-policy and on-policy training, is architecture-agnostic and will remain effective for future generations of models. Due to the limited response period, we would like to include them in the final version.
>
> Thank you very much for your suggestion.

---

> ### Author Response · Authors · 2025-11-22
> **Response to Reviewer pEZ4 (3/3)**
>
> > Q4: The method of smoothly combining with SFT should be generalizable to non-verifable task as well (and arguably more useful since SFT data could help constrain exploration space to prefered style/format). Any thoughts on such setting?
>
> Thank you very much for your helpful suggestions!
> We have conducted additional experiments on a non-verifiable task. We train Qwen2.5-7B-Instruct on the RaR-Medicine dataset [Ref1], a medical question-answering task without deterministic verification. The experimental results are shown below.
>
> | Methods | Score | Length |
> |---------|-------|--------
> | Expert Data | - | 550 |
> | Original Model | 67.5 | 402 |
> | Pure RL | 76.8 | 685 |
> | CHORD-$\mu$ | 80.6 | 1395 |
> | CHORD-$\phi$ | **81.3** | 1128 |
>
> These results demonstrate that **CHORD-$\mu$ and CHORD-$\phi$ significantly outperform pure RL**, further confirming the effectiveness of CHORD on non-verifiable tasks.
> The reward curves (please refer to Figure 10 in Appendix B.4 of the revised paper) show that CHORD-$\mu$ exhibits a "shift-readapt" pattern, while CHORD-$\phi$ achieves faster convergence and higher rewards compared to pure RL. These observations are consistent with those in verifiable tasks.
>
> We have added the above experimental results, analysis, and implementation details to Section 4.4 and Appendix B.4 in the revised paper. Thank you again for your helpful suggestions to further improve our paper!
>
> [Ref1] Rubrics as Rewards: Reinforcement Learning Beyond Verifiable Domains.
>
> -----------------
> Thank you for the positive feedback and insightful suggestions provided. We have uploaded **a revised paper that includes all the experiments in the above responses**, with the major modifications clearly highlighted. We sincerely appreciate your valuable feedback, which helped us to enhance the quality of our paper. Thank you!

---

> > ### Comment · Reviewer_pEZ4 · 2025-11-26
> > **Reviewer Response**
> >
> > Thanks for providing clarification and additional experiments. I will maintain my scoring.

---

> > > ### Author Response · Authors · 2025-11-27
> > > **Thank you for your reply**
> > >
> > > Thank you for your response and for acknowledging our submission with positive feedback!  We are pleased that our responses have addressed your concerns, and would be happy to continue the discussion and address any further questions.
> > >
> > > Thank you again!

---

### Official Review · Reviewer_UX1j · 2025-10-30

**Soundness:** 3
**Presentation:** 3
**Contribution:** 2
**Rating:** 6
**Confidence:** 3

**Summary:**

The paper provides a method to unify supervised fine-tuning (SFT) and reinforcement learning (RL) into a composite objective that weights the two individual objectives with a dynamic weighting value. Furthermore, the paper proposes to weight individual tokens to avoid issues that can arise from over-reliance on off-policy data and to encourage on-policy exploration. The effectiveness of these two weighting factors are studied empirically on Math and Tool-use datasets. The paper uses datasets provided by an expert model to learn policy. Empirical results suggest that the proposed method exceeds several reasonable baselines that includes both SFT and RL variants as well as  recently proposed works (LUFFY, SASR).

**Strengths:**

- The paper provides a clear description of the two objectives used in their method called CHORD. Furthermore, the paper clearly describes the experimental setup and results.
- The analysis conducted to motivate the method is clear.
- Empirical results suggest that the proposed method (CHORD-\phi) improves over several reasonable baselines on Math and Tool-use cases. The ablations included in the main paper suggest that the transition from offline imitation to online-RL learning is effective as it allows for exploration by the policy.

**Weaknesses:**

- The paper proposes one simple way to combine the two objectives. It's not clear why a convex combination of SFT and RL objectives is the right approach. Would it be possible to have generic weights for SFT and RL and let the model and data decide their optimal values?

- The objective for Chord-\phi uses a weight that looks like the variance of a Bernoulli random variable. Just like above, is this the optimal value for this weight? Are there any insights on what might happen if the base model is not as strong as the one considered in the experiments?

- Related to above, the analysis is conducted on Qwen2.5 for Math and Llama-3.2 for Tool use. What are the reasons for using these models the way they were used? Would the findings translate to other/newer models released in the (near) future?

**Questions:**

(repeated from weaknesses)

- Would it be possible to have generic weights for SFT and RL and let the model and data decide their optimal values?
- Are there any insights on what might happen if the base model is not as strong as the one considered in the experiments for the Chord-\phi's token weight?
- Are the findings applicable to newer models, especially architectures like MoEs that have been released or will be released in the future.

I would like to discuss the first two questions with the authors during rebuttal. The third question is asked to help the paper make generic but does not require a response.

---

> ### Author Response · Authors · 2025-11-22
> **Response to Reviewer UX1j (1/3)**
>
> We sincerely appreciate your detailed comments and valuable suggestions! We provide the following responses to address your concerns and answer your questions point by point.
>
> > W1 & Q1: It's not clear why a convex combination of SFT and RL objectives is the right approach. Would it be possible to have generic weights for SFT and RL and let the model and data decide their optimal values?
>
>
> (i) Firstly, at the global coefficient level, we allow $\mu$ to use different schedules and strategies, such as the decay schedule. As a natural and promising extension of our study, we further implemented a strategy where $\mu$ is adjusted based on model performance (i.e., reward) as you suggested, and conducted additional experiments.
> Specifically, for a given reward threshold $\tau$, the new $\mu$ is calculated as $\mu' = \max(0, \tau - reward\\_mean)$. The experimental results are shown below.
>
> | Method                            | Live  | Non-live | Overall |
> | :---                              | :---: | :---:    | :---:   |
> | Pure RL                           | 68.5  | 88.8     | 77.1    |
> | SFT-best                          | 59.2  | 84.2     | 69.8    |
> | SFT-best + RL                     | 67.6  | 87.9     | 76.1    |
> | CHORD-$\mu$ (fixed decay schedule)| 69.4  | 88.6     | 77.6    |
> | CHORD-$\mu$ (adaptive $\tau=0.5$) | 69.7  | 89.4     | 78.1    |
> | CHORD-$\mu$ (adaptive $\tau=0.7$) | 65.9  | 88.6     | 75.6    |
>
> From these results, we can observe that CHORD-$\mu$, when paired with various suitable strategies and hyperparameters, consistently achieves significant improvements over the baselines. We also find that the adaptive $\mu$ strategy can be sensitive to the parameter $\tau$, where an inappropriate hyperparameter can lead to performance degradation  (e.g., at $\tau=0.7$).
> Such results also support our motivation for balancing learning from SFT and RL at the token-wise level.
>
> (ii) Secondly, at the token-wise level, the weighting function $\phi$ effectively acts as a mechanism where the "model and data decide their optimal values" at a finer granularity.
>
> Recall that $\phi(y_t^*; \pi_\theta) = p_t(1-p_t)$, where $p_t$ is the model's probability of generating the expert token. This function is fully adaptive during training:
> - *Data-Driven*: It depends on the specific expert token $y_t^*$.
> - *Model-Driven*: It depends on the model's current confidence $p_t$.
>
> When the model is already confident ($p_t \approx 1$) or completely divergent ($p_t \approx 0$), the weight naturally becomes zero, reducing the SFT influence. This ensures that the optimal weight is dynamically determined for every single token without manual intervention.
>
> We have added the above experimental results to Section 4.3 and Appendix B.1 in the revised paper. We hope that these additional experimental results and discussions can further address the reviewer's question.

---

> ### Author Response · Authors · 2025-11-22
> **Response to Reviewer UX1j (2/3)**
>
> > W2: The objective for Chord-\phi uses a weight that looks like the variance of a Bernoulli random variable. Just like above, is this the optimal value for this weight?
>
> Thank you for your comments regarding $\phi(p)$. As we discussed above, the $\phi$ function decides the token weight adaptively by avoiding learning already confident or completely divergent tokens. We now conducted additional experiments to compare the weighting function $\phi(\cdot) = p∗(1−p)$ with the other functions, including:
> - *Entropy-Top*: Trains only on the top 5% highest-entropy tokens (i.e., entropy-based, token clipping);
> - *Entropy-Norm*: Normalizes SFT loss weights by entropy magnitude (i.e., entropy-based, continuous);
> - *IS Clip*: Importance sampling and gradient clipping for $p_t > 0.4$ (i.e., gradient clipping);
> - *Focal Loss*: Adapts focal loss: $(1-p)^\gamma$ (i.e., aggressive low-probability weighting).
>
> We show the experimental results below.
>
> On Tool-Use Task (we set rollouts per prompt to 8 and report the results after 100 training steps):
>
> | Method           | Live  | Non-live | Overall |
> | :---             | :---  | :---     | :---    |
> | Pure RL          | 68.5  | 88.8     | 77.1    |
> | Entropy-Top  | 69.1  | 89.4     | 77.8    |
> | Entropy-Norm     | 69.6  | 89.4     | 78.0    |
> | IS Clip          | 66.2  | 89.1     | 75.9    |
> | Focal Loss       | 65.6  | 84.0     | 73.4    |
> | $p∗(1−p)$ (ours) | **69.9** | **90.2** | **78.5** |
>
> On Mathematical Reasoning Tasks (we set rollouts per prompt to 16 and report the results after 400 training steps):
>
> | Method           | AMC23 | AIME2024 | AIME2025 |
> | :---             | :---  | :---     | :---     |
> | Pure RL          | 55.0  | 12.6     | 7.3      |
> | Entropy-Top  | 58.8  | **17.2** | 13.8     |
> | Entropy-Norm     | 52.5  | 15.0     | 9.2      |
> | IS Clip          | 55.0  | 13.9     | 12.0     |
> | Focal Loss       | 34.4  | 3.8      | 4.2      |
> | $p∗(1−p)$ (ours) | **59.7** | 14.0     | **14.2** |
>
>
> The experimental results confirm that **the proposed $\phi(\cdot)$ is an effective and robust instantiation of the principle of down-weighting tokens at both probability extremes**, providing consistent and significant gains over baselines.
>
> Beyond the proposed $\phi(\cdot) = p∗(1−p)$ (which serves as a concrete and interpretable instantiation), we also provide the general principle for instantiating advanced weighting functions for stable and selective learning from off-policy data. Please refer to the last paragraph in Section 4.3.
>
> We have added the above experimental results, analysis, and implementation details to Section 4.3 and Appendix B.2 in the revised paper. We hope these modifications and additional experiments can address the reviewer's question.

---

> ### Author Response · Authors · 2025-11-22
> **Response to Reviewer UX1j (3/3)**
>
> > Q2: Are there any insights on what might happen if the base model is not as strong as the one considered in the experiments for the Chord-\phi's token weight?
>
> Thank you for your insightful comments.
>
> Intuitively, for a weaker model, learning from expert data (SFT) would become more critical than on-policy learning (RL), since the diversity and effectiveness of on-policy exploration can be limited by the ability of the policy model. However, experimental results show that a weaker model has similar difficulty in absorbing knowledge from expert data.
>
> Specifically, we conducted experiments on a weaker model, Qwen2.5-3B-Instruct (we set rollouts per prompt to 16 and report after 400 training steps), for the mathematical reasoning tasks. The results are shown below.
>
> | Method | AMC23 | AIME2024 | AIME2025 |
> | :--- | :--- | :--- | :--- |
> | Original Model (3B) | 33.1 | 4.8 | 1.6 |
> | SFT | 22.5 | 1.9 | 1.0 |
> | Pure RL | 39.1 | 7.0 | 2.4 |
> | SFT+RL| 36.2  |6.2 | 3.7 |
> | CHORD-$\phi$ (ours) | **41.9** | **7.9** | **4.0** |
>
> From these results, we can observe:
> - While a larger model (Qwen2.5-7B-Instruct) can often "re-adapt" and improve with a modest amount of SFT data, the weaker model (Qwen2.5-3B-Instruct) might fail. Using the same SFT data, the weaker model experiences a performance drop. This suggests that weaker models may require unified approaches that combine SFT and RL even more than stronger models.
> - CHORD-$\phi$ consistently outperforms the baselines when applied to a weaker model, confirming that our principle of balancing exploration (uncertain tokens) and exploitation (high-confidence tokens) remains effective for enhancing model performance.
>
> We have added the above experimental results and discussions to Section 4.4 and Appendix B.5 in the revised paper. Thank you again for your insightful comments to improve our paper!
>
> > W3 & Q3: Are the findings applicable to newer models, especially architectures like MoEs that have been released or will be released in the future.
>
> We appreciate the reviewer's insightful question regarding the scalability of our findings to the latest models. We believe that our method, which targets a fundamental mechanism in harmonizing off-policy and on-policy training, is architecture-agnostic and will remain effective for future generations of models. Regarding the experiments with the MoE model, due to the limited response period, we would like to include them in the final version.
>
> Thank you very much for your suggestion.
>
> ----------------
>
> We are grateful for the constructive feedback provided by the reviewer. We have uploaded **a revised paper that includes all the experiments and discussions in the above responses**, with the major modifications clearly highlighted. We believe that this submission has been further improved based on your suggestions. We believe our response can address your questions, and we would be delighted to engage if you wish for further discussion.

---

> > ### Comment · Reviewer_UX1j · 2025-11-27
> > **Thanks**
> >
> > I thank the authors for their responses to my questions. I am satisfied with the answers. I continue to lean towards an accept which is reflected via the Rating of 6. I may revisit this score at the conclusion of the discussion period as it gives me a chance to follow author-reviewers discussions for this paper.

---

> > > ### Author Response · Authors · 2025-11-27
> > > **Thank you for your reply**
> > >
> > > We are glad to hear that you are satisfied with our responses and continue to lean towards accepting our paper.
> > >
> > > We sincerely appreciate your willingness to engage more deeply in the author-review discussion. Please allow us to briefly summarize the main revisions and improvements we have made in our response, so that you can easily access the key information at your convenience:
> > > - We explore various weighting designs, such as various token-wise weight $\phi$ and adaptive $\mu$. Please refer to Section 4.3 and Appendix B.1/B.2.
> > > - We expand our evaluation to include different experts (stylistically similar), policy models, and tasks (non-verifiable task). Please refer to Section 4.4 and Appendix B.3/B.4/B.5.
> > > - We highlight the novelty and contributions of CHORD. Please refer to Section 5 and Appendix C.2/D.1/D.4.
> > >
> > > ---
> > >
> > > We believe that the revised paper, which has been uploaded with all improvements clearly marked, has been further enhanced accordingly. We sincerely hope that these revisions will strengthen your continued support for its acceptance.
> > >
> > > Thank you very much for your insightful comments and helpful suggestions for further improving our submission. We would be happy to continue the discussion and address any further questions.

---

> > > ### Author Response · Authors · 2025-12-02
> > > **Update Additional Experiments on a MoE Model**
> > >
> > > Following up on our previous response, we have now completed the experiments on a Mixture of Experts (MoE) architecture model. We provide an update with new results from training **Phi-mini-MoE-instruct**, a lightweight MoE model with 3.8B total parameters and 1.1B active parameters.
> > >
> > > The experimental results on the Tool-Use task are presented below. (we set rollouts per prompt to 8 and report the results after 100 training steps)
> > >
> > > | Method | Live | Non-live | Overall |
> > > | :--- | :--- | :--- | :--- |
> > > | Original Model | 42.3 | 59.2 | 49.5 |
> > > | Pure RL | 51.2 | 69.3 | 58.9 |
> > > | SFT | 18.2 | 45.0 | 29.6 |
> > > | SFT + RL | 44.1 | 63.0 | 52.1 |
> > > | **CHORD-$\phi$ (ours)** | **52.1** | **74.5** | **61.6** |
> > >
> > > These results confirm that our method is architecture-agnostic, demonstrating its effectiveness on MoE models. This experiment validates that the benefits of our approach extend effectively beyond the previously tested Qwen and LLaMA models.
> > >
> > > We would like to thank the reviewer again for their constructive suggestion, which has helped us improve the quality and scope of our work.

---

### Official Review · Reviewer_f4d2 · 2025-10-30

**Soundness:** 3
**Presentation:** 2
**Contribution:** 2
**Rating:** 4
**Confidence:** 3

**Summary:**

In this paper, the authors propose CHORD, a unified framework that integrates Supervised Fine-Tuning (SFT) and Reinforcement Learning (RL) through a dynamically weighted objective. CHORD introduces two main components: a global coefficient μ that decays over time to balance imitation and exploration, and a token-wise weighting function ϕ(p)=p(1−p) to stabilize off-policy updates by emphasizing uncertain tokens. The method aims to mitigate instability when combining on-policy RL signals with off-policy expert data. Experiments on mathematical reasoning (OpenR1-Math) and tool-use tasks (ToolAce) demonstrate improved stability and modest gains over SFT→RL and recent hybrid methods such as LUFFY and SASR.

**Strengths:**

- The paper targets an important and timely problem in large language model post-training: how to combine supervised expert data with reinforcement learning in an effective way.


- The proposed framework is simple, well-motivated, and easy to implement in existing RLHF pipelines. The dual-control design (μ and ϕ) provides both stage-level and token-level balance between on- and off-policy learning.


- Experiments are extensive and include ablations (fixed vs. dynamic μ, with vs. without ϕ), entropy/reward analyses, and qualitative case studies that support the claimed stability improvements.

**Weaknesses:**

The novelty of CHORD is limited. The method reweights two existing loss terms (SFT and RL) using a dynamic coefficient and a heuristic token-wise weighting. Similar annealing strategies and uncertainty-based regularization have been explored in LUFFY, SRFT, and PPO variants with KL or imitation penalties.

- The token-level weighting ϕ(p)=p(1−p) is conceptually similar to entropy-based weighting and lacks theoretical justification for its specific form.

- The improvement margins over baselines are modest, and the experiments do not cover diverse post-training domains such as instruction-following or dialogue, leaving generality uncertain.

- The framework lacks a formal connection to off-policy correction theory or mixed-policy optimization, making it primarily heuristic rather than theoretically grounded.

**Questions:**

- Can the authors clarify how CHORD differs algorithmically from LUFFY or SRFT, beyond changing the weighting coefficients?


- Is there any theoretical interpretation (e.g., weighted policy gradient under mixed distributions) that supports the design of μ and ϕ?


- Have the authors tried learning μ adaptively (e.g., via reward variance or gradient norms) rather than fixing a decay schedule?


- Does the token-level ϕ weighting introduce significant computational overhead?


- How sensitive is the model’s stability to the exact shape of ϕ(p)? Would other functions (e.g., entropy-based) work similarly?

---

> ### Author Response · Authors · 2025-11-22
> **Response to Reviewer f4d2 (1/4)**
>
> We sincerely appreciate your detailed comments and valuable suggestions! We provide the following responses to address your concerns and answer your questions point by point.
>
> > W1: Similar annealing strategies and uncertainty-based regularization have been explored in LUFFY, SRFT, and PPO variants with KL or imitation penalties.
> >
> > Q1: Can the authors clarify how CHORD differs algorithmically from LUFFY or SRFT, beyond changing the weighting coefficients?
>
> Thank you for this constructive question.
>
> (i) One of the main contributions of this study lies in the framework that integrates SFT and RL in post-tuning, in which the global control component of $\mu$ serves as one component. The annealing strategies for $\mu$ represent only one instantiation and support more advanced implementations, as discussed in Section 3.2. This includes adaptive methods, as you suggested, and we provide corresponding comparisons and discussions among different instantiations of $\mu$; please refer to our response to Q3. These discussions and experimental results have been added to Section 4.3 and Appendix B.1 in the revised paper.
>
> (ii) Regarding the token-wise weighting function (i.e., the uncertainty-based regularization), CHORD is algorithmically distinct from LUFFY and SRFT. Specifically, CHORD is designed for a practical post-tuning scenario, where an instruction model serves as the starting point for further fine-tuning. Introducing off-policy expert data in this context results in a severe distributional mismatch, which leads to the “Shift-Readapt-Overfit” instability (see Fig. 2). Effectively combining SFT and RL in this setting requires selective absorption to prevent disruption of the model's established interaction patterns. CHORD is specifically tailored to stabilize this process, which is a key difference in both motivation and approach compared to the "Zero-RL" paradigm (tuning from base models, primarily targeted by SRFT and LUFFY). In the "Zero-RL" context, the goal is to instill reasoning capabilities into a model without a fixed response pattern; in contrast, our scenario demands enhanced stability for already-instructed models.
>
> Based on this distinct motivation, CHORD decouples expert and on-policy data streams to enable flexible incorporation of expert data. In SRFT and LUFFY, these streams are coupled by treating expert demonstrations as additional rollouts within the RL objective, which in turn requires strict one-to-one alignment between expert data and RL prompts. CHORD, on the other hand, applies the SFT objective exclusively to expert data and the RL objective to on-policy rollouts. This structural independence eliminates the need for rigid alignment and enables dynamic, flexible control over the use of expert data.
>
> Regarding the weighting mechanism, SRFT applies sample-level SFT loss weighting based on entropy, which results in the model learning full expert trajectories and can cause significant policy shifts. LUFFY up-weights low-probability tokens via importance sampling to promote exploration, which may destabilize instruction models that already possess a learned policy. In contrast, CHORD down-weights both highly confident (mastered) and improbable (disruptive) tokens, thereby maintaining stability and ensuring a smooth, dynamic balance between learning from expert data and the model’s own exploration at a finer granularity. We have also conducted additional experiments comparing SRFT/LUFFY in the tool-use task, showing that these methods designed for "Zero-RL" settings could fail when tuning an instruction-tuned model. The results and discussion have been added to Appendix D.1 in the revised paper.
>
> We have conducted experiments comparing CHORD with entropy-based methods; please refer to our response to Q5. These discussions and experimental results have been added to Section 4.4 and Appendix B.2 in the revised paper.
>
> Hope these responses address your concerns. Thank you again for your helpful suggestions to further improve our paper!

---

> ### Author Response · Authors · 2025-11-22
> **Response to Reviewer f4d2 (2/4)**
>
> > W2: The token-level weighting $\phi(p)=p(1−p)$ is conceptually similar to entropy-based weighting and lacks theoretical justification for its specific form.
> >
> > Q2: Is there any theoretical interpretation (e.g., weighted policy gradient under mixed distributions) that supports the design of $\mu$ and $\phi$?
> >
> > W4: The framework lacks a formal connection to off-policy correction theory or mixed-policy optimization, making it primarily heuristic rather than theoretically grounded.
>
> Thank you for your valuable comments and suggestions regarding the theoretical interpretation of CHORD. We provide the following discussions regarding how CHORD is theoretically motivated.
>
> The proposed CHORD is a principled and problem-driven method designed to address the empirical instabilities observed when integrating off-policy expert data with an already proficient instruction-tuned model, as illustrated by the "shift-readapt-overfit" phenomenon shown in Fig. 2. The theoretical inspiration of CHORD is summarized as follows:
>
> (i) **The decay schedule of $\mu$ is theoretically connected to the mitigation of exposure bias** [Ref1][Ref2], where models trained solely via teacher forcing (off-policy SFT) fail to generalize to their own generated distributions during inference, as discussed in Section 3.2. Scheduled sampling [Ref3] addresses this issue by dynamically sampling and mixing ground-truth tokens with model-generated tokens during training. CHORD generalizes this principle to the loss landscape. We conceptualize the SFT loss as the teacher forcing component (correcting via expert distribution) and the on-policy RL loss as the autoregressive component (optimizing the model's own distribution).
> The global coefficient $\mu$ acts as a continuous relaxation of the mixing probability in scheduled sampling. By annealing $\mu$, CHORD enforces a smooth, curriculum-based transition from off-policy correction to on-policy exploration, bridging the gap between training and inference to enable superior performance.
>
> (ii) **The token-wise weight $\phi(p)$ is introduced in response to the limitations of standard importance sampling in this context**. We observe that directly applying standard importance sampling (Eq. 4) in this setting tends to disproportionately amplify updates for high-probability tokens, leading to overfitting on those tokens and causing premature entropy collapse (as evidenced in Fig. 5). Consequently, rather than pursuing a strictly unbiased estimation, we design our correction term to act as a regularizer that prioritizes training stability. Furthermore, our proposed $\phi(p)$ presents an effective alternative with an information-theoretic interpretation: $p(1-p)$ quantifies the model’s uncertainty for a given token (as discussed in Sec. 4.3).
> By incorporating this measure as a weighting function, the learning process prioritizes tokens where the model exhibits the highest uncertainty, creating a favorable region for effective learning. This design serves two stabilizing roles:
> - It down-weights low-probability tokens ($p \to 0$), thereby preventing policy disruption, similar to importance sampling (IS).
> - It also down-weights high-probability tokens ($p \to 1$), averting overconfidence and entropy collapse as observed with IS, thus preserving exploration.
>
> Note that it is challenging to establish formal theoretical guarantees and could be a promising direction for future research, especially when lacking direct access to the expert policy distribution. Nonetheless, our proposed approach is both theoretically inspired and empirically validated. The observed improvements in performance and stable training dynamics provide strong evidence for the effectiveness of our design.
>
> We have added the above discussions to Appendix D.4 in the revised paper. We hope these responses address your concerns. Thank you again!
>
> [Ref1] Bridging the Gap Between Training and Inference for Neural Machine Translation. ACL 2019.
> [Ref2] Generalization in Generation: A Closer Look at Exposure Bias. ACL 2019.
> [Ref3] Scheduled Sampling for Sequence Prediction with Recurrent Neural Networks. Neurips 2015.

---

> ### Author Response · Authors · 2025-11-22
> **Response to Reviewer f4d2 (3/4)**
>
> > W3: The improvement margins over baselines are modest, and the experiments do not cover diverse post-training domains such as instruction-following or dialogue, leaving generality uncertain.
>
> (i) We would like to highlight that we carefully optimize the hyperparameters for SFT+RL baselines to ensure that the transition from SFT to RL was well-tuned for the adopted datasets. This process is resource-intensive, but it makes the performance of SFT+RL represent a top-level performance achievable through a hard switch between SFT and RL. CHORD-$\mu$ and CHORD-$\phi$ consistently outperform the baselines. Building on these results, our proposed method consistently achieves notable improvements across different tasks and models, with significant gains over the strong SFT+RT baselines, demonstrating the advances and effectiveness of our approach in unifying SFT and RL.
>
> (ii) We have conducted additional experiments on a non-verifiable task. We train Qwen2.5-7B-Instruct on the RaR-Medicine dataset [Ref1], a medical question-answering task without deterministic verification. The experimental results are shown below.
>
> | Methods | Score | Length |
> |---------|-------|--------
> | Expert Data | - | 550 |
> | Original Model | 67.5 | 402 |
> | Pure RL | 76.8 | 685 |
> | CHORD-$\mu$ | 80.6 | 1395 |
> | CHORD-$\phi$ | **81.3** | 1128 |
>
> These results demonstrate that CHORD-$\mu$ and CHORD-$\phi$ significantly outperform pure RL, further confirming the effectiveness of CHORD on non-verifiable tasks.
> The reward curves (please refer to Figure 10 in Appendix B.4 of the revised paper) show that CHORD-$\mu$ exhibits a "shift-readapt" pattern, while CHORD-$\phi$ achieves faster convergence and higher rewards compared to pure RL. These observations are consistent with those in verifiable tasks.
>
> We have added the above experimental results, analysis, and implementation details to Section 4.4 and Appendix B.4 in the revised paper. Thank you again for your helpful suggestions to further improve our paper!
>
> [Ref1] Rubrics as Rewards: Reinforcement Learning Beyond Verifiable Domains.
>
> > Q3: Have the authors tried learning $\mu$ adaptively (e.g., via reward variance or gradient norms) rather than fixing a decay schedule?
>
> Thank you for the insightful suggestions of learning $\mu$ adaptively.
>
> As a natural and promising extension of our study, we conducted additional experiments to explore a dynamic adjustment strategy of $\mu$ via reward variance. We defined a target threshold $\tau$ and calculated the coefficient as $\mu' = \max(0, \tau - reward\\_mean)$, suggesting that as the model improves (i.e., rewards increase), the SFT weight is naturally phased out.
>
> The experimental results on tool-use tasks are shown below.
> | Method | Live | Non-live | Overall |
> | :--- | :---: | :---: | :---: |
> | Original Model | 50.9 |39.9| 46.2|
> | Pure RL | 68.5 | 88.8 | 77.1 |
> | SFT-best | 59.2 | 84.2 | 69.8 |
> | SFT-best + RL | 67.6 | 87.9 | 76.1 |
> | CHORD-$\mu$ (fixing decay schedule) | 69.4 | 88.6 | 77.6 |
> | Adaptive $\mu$ ($\tau=0.5$) | 69.7 | 89.4 | 78.1 |
> | Adaptive $\mu$ ($\tau=0.7$) | 65.9 | 88.6 | 75.6 |
>
> From these results, we can observe that CHORD-$\mu$, when paired with various suitable strategies and hyperparameters, consistently achieves significant improvements over the baselines. We also find that the adaptive $\mu$ strategy can be sensitive to the parameter $\tau$, where an inappropriate hyperparameter can lead to performance degradation  (e.g., at $\tau=0.7$). Nevertheless, it is a promising direction to develop methods that learn $\mu$ adaptively, and we hope our experiments can further open up promising directions for future research.
>
> We have added the above experimental results, analysis, and implementation details to Section 4.3 and Appendix B.1 in the revised paper. Thank you again for your helpful suggestions regarding the design of the weighting scheme.

---

> ### Author Response · Authors · 2025-11-22
> **Response to Reviewer f4d2 (4/4)**
>
> > Q4: Does the token-level $\phi$ weighting introduce significant computational overhead?
>
> Thank you for your question.
>
> The token-level $\phi$ weighting does not introduce significant computational overhead. Specifically, the probabilities $p$ required for $\phi = p(1-p)$ are already computed as part of the standard SFT loss, so no extra forward pass is needed. We also detach the gradients for these weights. The only additional operation is a simple, element-wise multiplication, which incurs negligible cost compared to the overall training computation.
>
> We have added the above remark regarding the computational cost of $\phi$ in the revised paper (Section 4.3) accordingly. Thank you again!
>
>
> > Q5: How sensitive is the model’s stability to the exact shape of $\phi(p)$? Would other functions (e.g., entropy-based) work similarly?
>
> Thank you for your comments regarding the shape of $\phi(p)$.
>
> Here we compare the weighting function $\phi(\cdot) = p∗(1−p)$ with the other functions, including two entropy-based variants:
> - *Entropy-Top*: Trains only on the top 5% highest-entropy tokens (i.e., entropy-based variants, token clipping);
> - *Entropy-Norm*: Normalizes SFT loss weights by entropy magnitude (i.e., entropy-based variants, continuous);
> - *IS Clip*: Importance sampling and gradient clipping for $p_t > 0.4$ (i.e., margin, gradient clipping);
> - *Focal Loss*: Adapts focal loss: $(1-p)^\gamma$ (i.e., aggressive low-probability weighting).
>
> The experimental results are shown below.
>
> On Tool-Use Task (we set rollouts per prompt to 8 and report the results after 100 training steps):
>
> | Method           | Live  | Non-live | Overall |
> | :---             | :---  | :---     | :---    |
> | Pure RL          | 68.5  | 88.8     | 77.1    |
> | Entropy-Top  | 69.1  | 89.4     | 77.8    |
> | Entropy-Norm     | 69.6  | 89.4     | 78.0    |
> | IS Clip          | 66.2  | 89.1     | 75.9    |
> | Focal Loss       | 65.6  | 84.0     | 73.4    |
> | $p∗(1−p)$ (ours) | **69.9** | **90.2** | **78.5** |
>
> On Mathematical Reasoning Tasks (we set rollouts per prompt to 16 and report the results after 400 training steps):
>
> | Method           | AMC23 | AIME2024 | AIME2025 |
> | :---             | :---  | :---     | :---     |
> | Pure RL          | 55.0  | 12.6     | 7.3      |
> | Entropy-Top  | 58.8  | **17.2** | 13.8     |
> | Entropy-Norm     | 52.5  | 15.0     | 9.2      |
> | IS Clip          | 55.0  | 13.9     | 12.0     |
> | Focal Loss       | 34.4  | 3.8      | 4.2      |
> | $p∗(1−p)$ (ours) | **59.7** | 14.0     | **14.2** |
>
>
> The experimental results confirm that **the proposed $\phi(\cdot)$ is an effective and robust instantiation of the principle of down-weighting tokens at both probability extremes**, providing consistent and significant gains over baselines.
> The results also suggest that well-configured entropy-based variants (e.g., Entropy-Top), although the performance is not as consistent, may also work.
>
> Beyond the proposed $\phi(\cdot) = p∗(1−p)$ (which serves as a concrete and interpretable instantiation), we also provide the general principle for instantiating advanced weighting functions for stable and selective learning from off-policy data. Please refer to the last paragraph in Section 4.3.
>
> We have added the above experimental results, analysis, and implementation details to Section 4.3 and Appendix B.2 in the revised paper. Hope the above responses can address your concerns regarding the choice of $\phi(p)$.
>
> --------------
>
> We appreciate the reviewer's valuable feedback! We have uploaded **a revised paper that includes all the experiments and discussions in the above responses**, with the major modifications clearly highlighted. We believe that this submission has been further improved based on your suggestions. We hope these responses can address all your concerns and convince you to lean more towards acceptance of our paper.

---

### Official Review · Reviewer_hxFy · 2025-11-02

**Soundness:** 2
**Presentation:** 3
**Contribution:** 2
**Rating:** 4
**Confidence:** 3

**Summary:**

This paper examines why SFT+RL can underperform pure RL when expert data diverges from the policy, characterizing a “shift–readapt–overfit” dynamic. It proposes CHORD, which mixes GRPO with an auxiliary SFT loss: a global weight controls the overall expert influence; a token-wise weight emphasizes mid-probability tokens. Results on math and tool-use show improvements over several baselines.

**Strengths:**

1. This paper gives a clear empirical documentation of SFT instability under off-policy expert trajectories.

2. The hybrid objective (μ-weighted SFT + RL) is easy-to-implement.

3. The token-wise weighting is a simple stability heuristic; and the ablations on μ and training dynamics are decent.

**Weaknesses:**

1. The idea of combining supervised learning and RL during fine-tuning has been explored in prior works (e.g., SRFT, SimpleMix, LUFFY). CHORD uses a similar structure by optimizing a weighted sum of SFT loss and GRPO — with the addition of a global schedule μ and a token-level weight φ(y)=p(1–p).

2. The heuristic p(1−p) is plausible but lacks theoretical backing or strong comparisons to alternative uncertainty weights (entropy/focal/margin).

3. There is a heavy reliance on DeepSeek-R1 experts; and the analysis is limited for weaker/similar experts or different stylistic gaps.

**Questions:**

1. Can you provide controlled comparisons to SRFT/SimpleMix with matched compute/data and identical rollout settings? What is fundamentally new beyond weighting choices?

2. Can you compare the weighting funciton to alternatives (entropy, focal-style, margin/clipping) and report sensitivity?

3. Test experts that are weaker or stylistically closer to the policy; does μ decay still help, or does CHORD harm?

---

> ### Author Response · Authors · 2025-11-22
> **Response to Reviewer hxFy (1/3)**
>
> We sincerely appreciate your detailed comments and valuable suggestions! We provide the following responses to address your concerns and answer your questions point by point.
>
> > W1: The idea of combining supervised learning and RL during fine-tuning has been explored in prior works (e.g., SRFT, SimpleMix, LUFFY).
> >
> > Q1: Can you provide controlled comparisons to SRFT/SimpleMix with matched compute/data and identical rollout settings? What is fundamentally new beyond weighting choices?
>
> Thank you for this constructive question.
>
> (i) The combination of SFT and RL in post-tuning is indeed an important and promising direction for balancing learning from off-policy expert demonstrations and on-policy exploration. We have reviewed the differences in Section 5 and Appendix D.1, and highlight that there still remain challenges when integrating SFT and RL, which motivate our study.
>
> Specifically, we focus on a more practical post-tuning scenario, where an instruction model is used as the starting point for further post-tuning. Introducing off-policy expert data in this context results in a severe distributional mismatch, leading to the "Shift-Readapt-Overfit" instability (see Fig. 2). Combining SFT and RL in these scenarios requires selective absorption to avoid disrupting the model's established interaction patterns. CHORD is specifically designed to stabilize this process. This distinguishes our motivation and approach from the "Zero-RL" paradigm (tuning from base models, which is primarily targeted by SRFT and LUFFY), where the goal is to instill reasoning skills into a model without a fixed response pattern.
>
> Based on this different motivation, CHORD decouples data streams to facilitate flexible expert data incorporation. While SRFT and LUFFY couple these streams by treating expert demonstrations as additional rollouts within the RL objective, this approach requires a strict one-to-one alignment between expert data and RL prompts. In contrast, CHORD applies the SFT objective exclusively to expert data and the RL objective to on-policy rollouts. This structural independence eliminates the need for rigid alignment and enables flexible and dynamic control over expert data.
> For the weighting mechanism, SRFT weights the sample-level SFT loss based on entropy, which still leads the model to learn entire expert trajectories and cause a policy shift. LUFFY up-weights low-probability tokens on top of importance sampling weights to force exploration, which can destabilize instruction models with established policies. In contrast, CHORD down-weights confident (mastered) or improbable (disruptive) tokens to maintain stability, ensuring a smooth and dynamic balance between learning from expert data and the model’s own exploration at a finer granularity. SimpleMix, on the other hand, operates within a DPO framework, which requires preference-pair data $(y_{chosen}, y_{reject})$. Our method is GRPO-based and integrates single expert demonstrations $y^*$. Furthermore, SimpleMix combines off-policy and on-policy data via simple dataset-level sampling (e.g., a 50/50 mix), lacking dynamic weighting to manage the distributional shifts that are central to our problem setting.
>
>
> (ii) We conduct controlled comparisons among CHORD, SRFT, and LUFFY with matched computation, data, and identical rollout settings on the tool-use dataset. The experimental results confirm the consistent improvement brought by the proposed CHORD.
>
> | Method | Live | Non-live | Overall |
> |--------|------|----------|---------|
> | Pure RL | 68.5 | 88.8 | 77.1 |
> | SFT-best | 59.2 | 84.2 | 69.8 |
> | SFT-best + RL | 67.6 | 87.9 | 76.1 |
> | LUFFY | 67.2 | 88.0 | 76.1 |
> | SRFT | 64.6 | 85.8 | 73.6 |
> | CHORD-$\mu$ | **69.4** | **88.6** | **77.6** |
> | CHORD-$\phi$ | **69.9** | **90.2** | **78.5** |
>
> In summary, we believe that CHORD provides a novel contribution to the research area of combining SFT and RL. It identifies specific failure modes of off-policy learning for instruction models (instability and pattern disruption) and proposes a fine-grained weighting mechanism to address these challenges. The controlled comparisons with SRFT and LUFFY further demonstrate the effectiveness of CHORD.
>
> We have added the above experimental results and analysis to Appendix D.1 in the revised paper. Hope these responses address your concerns. Thank you again for your helpful suggestions to further improve our paper!

---

> ### Author Response · Authors · 2025-11-22
> **Response to Reviewer hxFy (2/3)**
>
> > W2: The heuristic p(1−p) is plausible but lacks theoretical backing or strong comparisons to alternative uncertainty weights (entropy/focal/margin).
> >
> > Q2: Can you compare the weighting function to alternatives (entropy, focal-style, margin/clipping) and report sensitivity?
>
> Thank you for your constructive suggestions.
>
> - Beyond the proposed $\phi(\cdot) = p(1−p)$ (which serves as a concrete and interpretable instantiation), we also provide the general principle for instantiating advanced weighting functions for stable and selective learning from off-policy data. Please refer to the last paragraph in Section 4.3.
> - We conducted additional experiments to compare the weighting function $\phi(\cdot) = p(1−p)$ with the alternatives you suggested (entropy, focal-style, margin/clipping):
>     - *Entropy-Top*: Trains only on the top 5% highest-entropy tokens (i.e., thresholding, hard clipping);
>     - *Entropy-Norm*: Normalizes SFT loss weights by entropy magnitude (i.e., continuous entropy);
>     - *IS Clip*: Importance sampling and gradient clipping for $p > 0.4$ (i.e., margin, gradient clipping);
>     - *Focal Loss*: Adapts focal loss $(1-p)^\gamma$ (i.e., aggressive weighting low-prob tokens).
>
> The experimental results are shown below.
>
> On Tool-Use Task (we set rollouts per prompt to 8 and report the results after 100 training steps):
>
> | Method           | Live  | Non-live | Overall |
> | :---             | :---  | :---     | :---    |
> | Pure RL          | 68.5  | 88.8     | 77.1    |
> | Entropy-Top  | 69.1  | 89.4     | 77.8    |
> | Entropy-Norm     | 69.6  | 89.4     | 78.0    |
> | IS Clip          | 66.2  | 89.1     | 75.9    |
> | Focal Loss       | 65.6  | 84.0     | 73.4    |
> | $p(1−p)$ (ours) | **69.9** | **90.2** | **78.5** |
>
> On Mathematical Reasoning Tasks (we set rollouts per prompt to 16 and report the results after 400 training steps):
>
> | Method           | AMC23 | AIME2024 | AIME2025 |
> | :---             | :---  | :---     | :---     |
> | Pure RL          | 55.0  | 12.6     | 7.3      |
> | Entropy-Top  | 58.8  | **17.2** | 13.8     |
> | Entropy-Norm     | 52.5  | 15.0     | 9.2      |
> | IS Clip          | 55.0  | 13.9     | 12.0     |
> | Focal Loss       | 34.4  | 3.8      | 4.2      |
> | $p(1−p)$ (ours) | **59.7** | 14.0     | **14.2** |
>
>
> The experimental results confirm that **the proposed $\phi(\cdot)$ is an effective and robust instantiation of the principle of down-weighting tokens at both probability extremes**, providing consistent gains over baselines.
>
> We have added the above experimental results, analysis, and implementation details to Section 4.3 and Appendix B.2 in the revised paper. Thank you again for your helpful suggestions regarding the comparison of weighting functions.

---

> ### Author Response · Authors · 2025-11-22
> **Response to Reviewer hxFy (3/3)**
>
> > W3: There is a heavy reliance on DeepSeek-R1 experts; and the analysis is limited for weaker/similar experts or different stylistic gaps.
> >
> > Q3: Test experts that are weaker or stylistically closer to the policy; does $\mu$ decay still help, or does CHORD harm?
>
> Thank you for your insightful suggestions.
> We have conducted additional experiments using Qwen2.5-72B-Instruct as an expert, which is weaker than DeepSeek-R1 and stylistically closer to our policy model LLaMA3.2-3B-Instruct. Several examples are provided in Appendix B.3, confirming that, compared to DeepSeek-R1 (which produces verbose, elaborate reasoning with extensive meta-commentary), Qwen2.5-72B-instruct adopts concise reasoning and a structured style that is close to LLaMA3.2-3B-Instruct.
>
> The experimental results on tool-use tasks when using Qwen2.5-72B-Instruct as the expert are shown below:
>
> | Method | Expert | Live | Non-live | Overall |
> |--------|---------------|------|----------|---------|
> | Pure RL | — | 68.5 | 88.8 | 77.1 |
> | SFT-best + RL | DeepSeek-R1 | 67.4 | 87.9 | 76.1 |
> | SFT-best + RL | Qwen2.5-72B | 68.1 | 90.6 | 77.7 |
> | CHORD-$\mu$ | DeepSeek-R1 | 69.4 | 88.6 | 77.6 |
> | CHORD-$\mu$ | Qwen2.5-72B | 68.8 | 90.6 | 78.1 |
> | CHORD-$\phi$ | DeepSeek-R1 | 69.9 | 90.2 | 78.5 |
> | CHORD-$\phi$ | Qwen2.5-72B | 68.9 | 90.9 | 78.3 |
>
> From these results, we observe the following insights:
> - **Consistent Improvement Across Experts**: Whether the adopted expert is DeepSeek-R1 or Qwen2.5-72B, CHORD-$\mu$ and CHORD-$\phi$ consistently improve over Pure RL and SFT-best+RL, confirming the effectiveness of CHORD in unifying learning from both (diverse) expert data and on-policy explorations.
> - **Impact of Response Pattern Shift**: We find methods that tend to mimic SFT patterns(e.g., SFT+RL and CHORD-$\mu$) can yield greater gains when the expert exhibits a more similar response pattern to the base policy. This aligns with our discussion in the paper, i.e., the effectiveness of unifying SFT and RL depends not only on the quality of expert data, but also on the degree of pattern shift it introduces relative to the base model’s existing policy.
>
> We have added the above experimental results and analysis to Section 4.4 and Appendix B.3 in the revised paper. Thank you again for your helpful suggestions regarding the comparison of different experts.
>
> -----------------
>
> We are grateful for the constructive feedback provided by the reviewer. We have taken great care in addressing the questions raised, and have uploaded **a revised paper that includes all the experiments in the above responses**, with the major modifications clearly highlighted. We believe that this submission has been further improved based on your suggestions. We hope these responses can address all your concerns and convince you to lean more towards acceptance of our paper.

---

### Author Response · Authors · 2025-11-27
**General Response & Welcoming Further Feedback**

Dear Reviewers,

We sincerely appreciate your time in providing detailed comments and helpful suggestions! We would like to summarize the common positive feedback and our main efforts during the rebuttal period.

---

First of all, we are grateful for the positive comments from the reviewers, acknowledging that our work features:

-   An important and timely topic that combines SFT and RL (from Reviewer f4d2).
-   Clear motivation and insightful analysis regarding the "shift–readapt–overfit" phenomenon (from Reviewers hxFy, UX1j, and pEZ4).
-   A well-motivated method design, praised for its clarity (from Reviewers hxFy, f4d2, UX1j, and pEZ4).
-   An easy-to-implement approach with demonstrated empirical improvements (from Reviewers hxFy, f4d2, UX1j, and pEZ4).
-   Comprehensive evaluation, including detailed empirical results and training dynamic ablations (from Reviewers hxFy, f4d2, UX1j, and pEZ4).
-   Clear presentation, with a clear description of the objectives, experimental setup, and results (from Reviewer UX1j).

---

During the rebuttal period, we have made every effort to provide detailed responses to each question. We have conducted additional experiments and included new analyses to address all concerns, with major updates highlighted in the revised paper. The key improvements are summarized as follows:


- **Exploring weighting designs**:
    - We explore an adaptive strategy for the global coefficient $\mu$ based on reward signals, demonstrating the flexibility of our framework (please refer to *Sec 4.3 and Appendix B.1*).
    - We conduct extensive experiments comparing our token-wise weight $\phi(p)$ with alternatives, including entropy-based methods, focal loss, and gradient clipping, demonstrating both the effectiveness and robustness of our design. We also discuss its negligible computational overhead (please refer to *Sec 4.3 and Appendix B.2*).


- **Expanding evaluation across different experts, policy models, and tasks**:
    - We test with a weaker and stylistically closer expert (Qwen2.5-72B-Instruct) to demonstrate CHORD's robustness across different expert data distributions (please refer to *Sec 4.4 and Appendix B.3*).
    - We evaluate CHORD on a non-verifiable task (Medical QA), confirming its applicability beyond domains with deterministic rewards (please refer to *Sec 4.4 and Appendix B.4*).
    - We apply CHORD to a smaller policy model (Qwen2.5-3B-Instruct), showing that SFT/RL synergy generalizes to less capable models (please refer to *Sec 4.4 and Appendix B.5*).


- **Highlighting the novelty and contribution of CHORD**:
    - We add further discussions and comprehensive comparisons with existing studies (e.g., *Simple Mix, SRFT, and LUFFY*), confirming the structural differences, algorithmic novelty, and empirical superiority of CHORD in post-tuning instruction models (please refer to *Section 5, Appendix C.2, and Appendix D.1*).
    - We provide an in-depth discussion on the theoretical motivations for CHORD, specifically its role in mitigating exposure bias and promoting stable off-policy regularization (please refer to *Appendix D.4*).

We have uploaded **a revised paper that includes all the experiments, analysis, and discussions** mentioned above. We trust that these updates have effectively addressed your concerns and clarified any potential misunderstandings. We believe the submission has been substantially strengthened thanks to your valuable suggestions.

---

We are eager to know whether our responses and revisions have met your expectations. Thank you once again for your invaluable time and guidance!

---

### Author Response · Authors · 2025-12-03
**Summary of Responses & Revisions to Area Chairs (1/2)**

Dear Area Chairs,

Thank you very much for managing the review process for our submission! To support your assessment, we have prepared a summary of our responses to the reviewers' questions and the corresponding improvements in the revised paper.

---

## Positive Feedback
Firstly, we are grateful for the positive comments from reviewers, acknowledging that our work features:
- An important and timely topic that combines SFT and RL (Reviewer f4d2).
- Clear motivation and insightful analysis regarding the "shift–readapt–overfit" phenomenon (Reviewers hxFy, UX1j, pEZ4).
- A well-motivated method design praised for its clarity and rationale (Reviewers hxFy, f4d2, UX1j, pEZ4).
- An easy-to-implement approach with demonstrated empirical improvements (Reviewers hxFy, f4d2, UX1j, pEZ4).
- Comprehensive evaluation, including detailed empirical results and training dynamic ablations (Reviewers hxFy, f4d2, UX1j, pEZ4).
- Clear presentation with a clear description of the objectives, experimental setup, and results (Reviewer UX1j).


## Responses and Revisions
We have organized the reviewers' feedback into major categories and made substantial revisions for each, incorporating new experiments and analyses as summarized below.

### 1. Experiments and Analysis regarding the Weighting Designs

 - 1.1 We explore an adaptive strategy for the global coefficient $\mu$ based on reward signals, demonstrating the flexibility of our framework beyond the decay schedule.
    * This responds to *Reviewer f4d2(Q3)* and *Reviewer UX1j(W1, Q1)*.
    * These revisions have been added to $\textcolor{blue}{\text{Sec 4.3}}$ and $\textcolor{blue}{\text{Appendix B.1}}$.

 - 1.2 We conduct extensive experiments comparing our token-wise weight $\phi(p)$ with alternatives, including entropy-based methods, focal loss, and gradient clipping, demonstrating both the effectiveness and robustness of our design.
    * This responds to *Reviewer hxFy(W2, Q2)*, *Reviewer f4d2(Q5)*, and *Reviewer UX1j(W2)*
    * These revisions have been added to $\textcolor{blue}{\text{Sec 4.3}}$ and $\textcolor{blue}{\text{Appendix B.2}}$.

### 2. Expanding Evaluation across Different Experts, Policy models, and Tasks

- 2.1 We conduct experiments with a weaker and stylistically closer expert (Qwen2.5-72B-Instruct) to demonstrate CHORD's robustness across different expert data distributions.
    * This responds to *Reviewer hxFy(W3, Q3)*
    * These revisions have been added to $\textcolor{blue}{\text{Sec 4.4}}$ and $\textcolor{blue}{\text{Appendix B.3}}$.

- 2.2 We evaluate CHORD on a non-verifiable task (Medical QA), confirming its utility and generalization beyond verifiable domains.
    * This responds to *Reviewer f4d2(W3)* and *Reviewer pEZ4(Q4)*
    * These revisions have been added to $\textcolor{blue}{\text{Sec 4.4}}$ and $\textcolor{blue}{\text{Appendix B.4}}$.

- 2.3 We apply CHORD to a smaller policy model (Qwen2.5-3B-Instruct), showing how SFT/RL synergy generalizes to less capable models.
    * This responds to *Reviewer UX1j(Q2)*
    * These revisions have been added to $\textcolor{blue}{\text{Sec 4.4}}$ and $\textcolor{blue}{\text{Appendix B.5}}$.

- 2.4 We apply CHORD to a Mixture of Experts (MoE) architecture model to demonstrate that our method is architecture-agnostic and its effectiveness extends beyond the previously tested dense models.
    * This responds to *Reviewer UX1j(W3, Q3)*.
    * These revisions have been added to $\textcolor{blue}{\text{Appendix B.6}}$.

- 2.5 We conduct further comparisons with LUFFY and SRFT on tool-use tasks, showing that CHORD achieves greater training stability and effectiveness in post-tuning instruction models.
    * This responds to *Reviewer hxFy(W1, Q1)* and *Reviewer f4d2(W1, Q1)*.
    * These revisions have been added to $\textcolor{blue}{\text{Appendix D.1}}$.

---

> ### Author Response · Authors · 2025-12-03
> **Summary of Responses & Revisions to Area Chairs (2/2)**
>
> ### 3. Further Discussions
>
> - 3.1 We provide a brief discussion regarding the negligible computational overhead of $\phi(p)$.
>     * This responds to *Reviewer f4d2(Q4)*
>     * These revisions have been added to $\textcolor{blue}{\text{Sec 4.3}}$.
>
> - 3.2 We add further discussions regarding the comparisons between CHORD and existing studies (e.g., *Simple Mix, SRFT, and LUFFY*), confirming the structural differences and algorithmic novelty of CHORD in post-tuning instruction models.
>     * This responds to *Reviewer hxFy(W1, Q1)* and *Reviewer f4d2(W1, Q1)*
>     * These revisions have been added to $\textcolor{blue}{\text{Sec 5}}$, $\textcolor{blue}{\text{Appendix C.2}}$, and $\textcolor{blue}{\text{Appendix D.1}}$.
>
> - 3.3 We provide an in-depth discussion on the theoretical motivations for CHORD, specifically its role in mitigating exposure bias and promoting stable off-policy regularization.
>     * This responds to *Reviewer f4d2(W2, W4, Q2)*
>     * These revisions have been added to $\textcolor{blue}{\text{Appendix D.4}}$.
>
> - 3.4 We add discussions regarding the difference in SFT/RL synergy across tasks and how this difference affects task-related performance.
>     * This responds to *Reviewer pEZ4(W1, Q1)*
>     * These revisions have been added to $\textcolor{blue}{\text{Sec.4.2}}$ and $\textcolor{blue}{\text{Appendix D.2}}$.
>
> - 3.5 We add more discussion on why scaling SFT is not enough and highlight the necessity of on-policy learning.
>     * This responds to *Reviewer pEZ4(Q2)*
>     * These revisions have been added to $\textcolor{blue}{\text{Appendix D.3}}$.
>
>
> ## Summary of Author-Reviewer Discussions
>
> We have taken great care to address the reviewers' concerns. During the author-reviewer discussion period, Reviewer UX1j and Reviewer pEZ4 acknowledged that our responses thoroughly addressed all of their concerns and gave positive feedback on our submission. For Reviewers hxFy and f4d2, while they did not further engage in the author-reviewer discussion, we are confident that our detailed, point-by-point response has thoroughly addressed the issues they raised.
>
> ---
> We would like to once again thank all the reviewers for their constructive feedback, which helped us strengthen the paper.
>
> We also deeply appreciate the AC's support and the considerable efforts invested in overseeing the reviews, especially given the current circumstances. We hope that our revised submission and consolidated responses will help you to fairly assess our work.
>
> Thank you again for your time and efforts in ensuring a constructive review process!

---

### Meta-Review · Area_Chair_7hst · 2026-01-06

**Summary:**

The paper proposes a combined objective unifying SFT and RL with dynamic weighting, plus token-level weighting to mitigate off-policy data over-reliance and boost on-policy exploration. On Math and Tool-use datasets, it outperforms SFT/RL baselines.

**Reviewer Concerns:**

Addressed concerns:
1. Exploring other weighting designs
2. Expanding evaluation across different experts, policy models, and tasks
3. Clarifying novelty and contribution of CHORD


There are no obvious outstanding concerns.

**Reviewer Scores:**

6, 6, 6, 6

---

### Decision · Program_Chairs · 2026-01-26

Accept (Poster)